# BEST RESPONSE SHAPING

## ABSTRACT

We investigate the challenge of multi-agent deep reinforcement learning in partially competitive environments, where traditional methods struggle to foster reciprocity-based cooperation. LOLA and POLA agents learn reciprocity-based cooperative policies by differentiation through a few look-ahead optimization steps of their opponent. However, there is a key limitation in these techniques. Because they consider a few optimization steps, a learning opponent that takes many steps to optimize its return may exploit them. In response, we introduce a novel approach, Best Response Shaping (BRS), which differentiates through an opponent approximating the best response, termed the "detective." To condition the detective on the agent's policy for complex games we propose a state-aware differentiable conditioning mechanism, facilitated by a question answering (QA) method that extracts a representation of the agent based on its behaviour on specific environment states. To empirically validate our method, we showcase its enhanced performance against a Monte Carlo Tree Search (MCTS) opponent, which serves as an approximation to the best response in the Coin Game. This work expands the applicability of multi-agent RL in partially competitive environments and provides a new pathway towards achieving improved social welfare in general sum games.

## 1 INTRODUCTION

Reinforcement Learning (RL) algorithms have enabled agents to perform well in complex high-dimensional games like Go (Silver et al., 2016) and StarCraft (Vinyals et al., 2019). The end goal of RL is to train agents that can help humans solve challenging problems. Inevitably, these agents will need to integrate in real-life scenarios that require interacting with humans and other learning agents. While multi-agent RL training shines in fully cooperative or fully competitive environments, it often fails to find reciprocity-based cooperation in partially competitive environments. One such example is the failure of multi-agent RL (MARL) agents to learn policies like tit-for-tat (TFT) in the Iterated Prisoner's Dilemma (IPD) (Foerster et al., 2018).

Despite the toy-ish character of common general-sum games such as IPD, these sorts of problems are ubiquitous in both society and nature. Consider a scenario where two countries (agents), strive to maximize their industrial output while also ensuring a suitable climate for production by limiting carbon emissions. On the one hand, each country (agent) would like to see the other country fulfill it's obligations to limit carbon emissions. Yet on the other hand, each one is motivated to emit more carbon themselves to achieve higher industrial yields. An effective climate treaty would compel each country – likely through the threat of penalties – to abide by the agreed limits to carbon emissions. If these agents fail to develop such tit-for-tat-like strategies they will likely converge to an unfortunate mutual escalation of consumption and carbon emission.

Foerster et al. (2018) proposed Learning with Opponent-Learning Awareness (LOLA), an algorithm that successfully learns TFT behavior in the IPD setting by differentiating through an assumed single naive gradient step taken by the opponent. Building upon this, Zhao et al. (2022) introduced proximal LOLA (POLA), which further enhances LOLA by assuming a proximal policy update for the opponent. This improvement allows for the training of Neural Network (NN) policies in more complex games, such as the Coin Game (Foerster et al., 2018). To the best of our knowledge, POLA is the only method that reliably trains reciprocity-based cooperative agents in the Coin Game.

Despite its success on the Coin Game, POLA has its limitations. While POLA is learning with opponent-learning awareness, its modeling of opponent learning is limited to a few look-ahead optimization steps. This renders POLA vulnerable to exploitation by opponents engaging in additional optimization. In particular, our analysis of POLA agents trained on the Coin Game demonstrates that POLA is susceptible to exploitation by the best response opponent. When the opponent is specifically trained to maximize its own return against a fixed policy trained by POLA, the first exploits the former. Also, this limitation can hinder POLA's scalability; it can't differentiate through all opponent optimization steps. This is particularly problematic if the opponent is a complex neural network, as many optimization steps are needed to approximate its learning.

In this paper, we present a novel approach called Best Response Shaping (BRS). Our method is based on the construction of an opponent that approximates the best response policy against a given agent. We refer to this opponent as the "detective." The overall concept is depicted in Figure 1: the detective undergoes training against agents sampled from a diverse distribution. To train the agent, we differentiate through the detective opponent. Unlike approaches such as LOLA and POLA, which assume few look-ahead optimization steps, our method relies on the detective issuing the best response to the current agent through policy conditioning.

The detective conditions on an embedding of the agent's policy that effectively captures its behavior across various states of the environment. Extracting such a representation is a non-trivial task (Harb et al., 2020). A straightforward approach of concatenating all policy parameters into a single representation results in a loss of architectural information and requires a large number of samples to be effective. Alternatively, conditioning the representation on the agent's behavior in specific query states, as done in Harb et al. (2020), can be attempted. However, learning these query states to enable generalization of the agent's behavior is, by itself, a difficult problem. To address this, we introduce a question-answering (QA) mechanism dependent on the current state of the environment, which serves as a means to extract a representation of the agent policy. The detective evaluates the agent's policy (answers) based on specific environment states (questions) given the current state.

We empirically validate our method on Iterated Prisoner's Dilemma (IPD) and the Coin Game. Given the dependency on the opponent's policy for an agent's outcomes, it is not always straightforward to evaluate and compare policies of different agents in games. This is especially true in non-zero-sum games that exhibit both cooperative and competitive aspects. In this paper, we advocate that a reasonable point of comparison is the agent's outcome when facing a best response opponent, which we approximate by Monte Carlo Tree Search (MCTS). We show that while the MCTS does not fully cooperate with POLA agents, they fully cooperate with our BRS agent.

**Main Contributions**. We summarize our main contributions below:

- We identify that the best response opponent, as approximated by Monte Carlo Tree Search (MCTS), does not cooperate with POLA agents. MCTS exploits the POLA agents achieving a higher return than it would through complete cooperation.
- To address this vulnerability, we introduce the BRS method, which trains an agent by differentiating through an opponent approximating the best response (referred to as the 'detective opponent'). We empirically validate our method and demonstrate that the best response to BRS agents is indeed full cooperation as shown in Figure 3.
- Additionally, we propose a state-aware differentiable conditioning mechanism for the detective opponent, enabling it to condition on the agent's policy.

## 2 BACKGROUND

### 2.1 MULTI AGENT REINFORCEMENT LEARNING

An $N$-agent Markov Games is denoted by a tuple $\left(N, \mathcal{S}, \left\{\mathcal{A}^i\right\}_{i=1}^N, \mathbb{P}, \left\{r^i\right\}_{i=1}^N, \gamma\right)$. Here, $N$ represents the number of agents, $\mathcal{S}$ the state space of the environment, and $\mathcal{A} := \mathcal{A}^1 \times \cdots \times \mathcal{A}^N$ the set of actions for each agent. Transition probabilities are denoted by $\mathbb{P} : \mathcal{S} \times \mathcal{A} \to \Delta(\mathcal{S})$ and the reward function by $r^i : \mathcal{S} \times \mathcal{A} \to \mathbb{R}$. Lastly, $\gamma \in [0, 1]$ is the discount factor. In a multi-agent reinforcement learning problem each agent attempts to maximize their return $R^i = \sum_{t=0}^{\infty} \gamma^t r_t^i$. The policy of agent $i$ is denoted by $\pi_{\theta_i}^i$ where $\theta_i$ are policy parameters. In Deep RL these policies are neural networks. These policies will be trained via gradient estimators such as REINFORCE (Sutton et al., 1999).

## 2.2 SOCIAL DILEMMAS AND THE ITERATED PRISONER'S DILEMMA

In the context of general sum games, social dilemmas emerge when individual agents striving to optimize their personal rewards inadvertently undermine the collective outcome or social welfare. This phenomenon is most distinct when the collective result is inferior to the outcome that could have been achieved through full cooperation. Theoretical studies, such as the Prisoner's Dilemma, illustrate scenarios where each participant, though individually better off confessing, collectively achieves a lower reward compared to remaining silent.

However, in the Iterated Prisoner's Dilemma (IPD), unconditional defection ceases to be the dominant strategy. For instance, against an opponent following a tit-for-tat (TFT) strategy, perpetual cooperation results in a higher return for the agent. It might be expected that MARL, designed to maximize each agent's return, would discover the TFT strategy, as it enhances both collective and individual returns, and provides no incentive for policy change, embodying a Nash Equilibrium. Yet, empirical observations reveal that standard RL agents, trained to maximize their own return, typically converge to unconditional defection.

This exemplifies one of the key challenges of multi-agent RL in general sum games: during training, agents often neglect the fact that other agents are also in the process of learning. To address this issue, and if social welfare is the primary consideration, one could share the rewards among the agents during training. For instance, training both agents in an IPD setup to maximize the collective return would lead to a constant cooperation. However, this approach is inadequate if the goal is to foster reciprocation-based cooperation. A policy is sought that incites the opponent to cooperate in order to maximize their own return. While TFT is one such policy, manually designing a similar TFT policies in other domains is neither desirable nor feasible, underscoring the necessity to develop novel training algorithms that can discover these policies.

## 3 RELATED WORK

LOLA Foerster et al. (2018) attempts to shape the opponent by taking the gradient of the value with respect to a one-step look ahead of the opponent's parameters. Instead of considering the expected return under the current policy parameter pair, $V^1(\theta_i^1, \theta_i^2)$, LOLA optimizes $V^1(\theta_i^1, \theta_i^2 + \Delta\theta_i^2)$ where $\Delta\theta_i^2$ denotes a naive learning step of the opponent. To make a gradient calculation of the update $\Delta\theta_i^2$, LOLA considers the surrogate value given by the first order Taylor approximation of $V^1(\theta_i^1, \theta_i^2 + \Delta\theta_i^2)$. Since for most games the exact value cannot be calculated analytically, the authors introduce a policy gradient formulation that relies on environment roll-outs to approximate it. This method is able to find tit-for-tat strategies on the Iterated Prisoner's Dilemma.

POLA Zhao et al. (2022) introduces an idealized version of LOLA that is invariant to policy parameterization. To do so, each player attempts to increase the probability of actions that lead to higher returns while penalizing the Kullback-Leibler divergence in policy space relative to their policies at the previous time step. Similar to the proximal point method, each step of POLA constitutes an optimization problem that is solved approximately through gradient descent. Like LOLA, POLA uses trajectory roll-outs to estimate the value of each player and applies the reinforce estimator to compute gradients. POLA effectively achieves non exploitable cooperation on the IPD and the Coin Game improving on the shortcomings of its predecessor.

Lu et al. (2022) considers a meta-game where at each meta-step a full game is played and the meta-reward is the return of that game. The agent is then a meta-policy that learns to influence the opponent's behaviour over these rollouts. M-FOS changes the game and is not comparable to our method which considers learning a single policy. Baker (2020) changes the structure of the game where each agent is sharing reward with other agents. The agents are aware of this grouping of rewards via a noisy version of the reward sharing matrix. In the test time, the representation matrix is set to no reward sharing and no noise is added to this matrix.

Some methods in the literature strive to add behavioral diversity to train strong agents for games using Determinantal Point Processes (DPPs) (Nieves et al., 2021; Yang et al., 2020). A DPP is a probabilistic method, commonly used in physics, to sample diverse variations from a ground set proportionally to a similarity metric. This technique resembles our agent replay buffer, which is intended to add behavioral diversity that is relatively close to the current policy.

Detective Training:

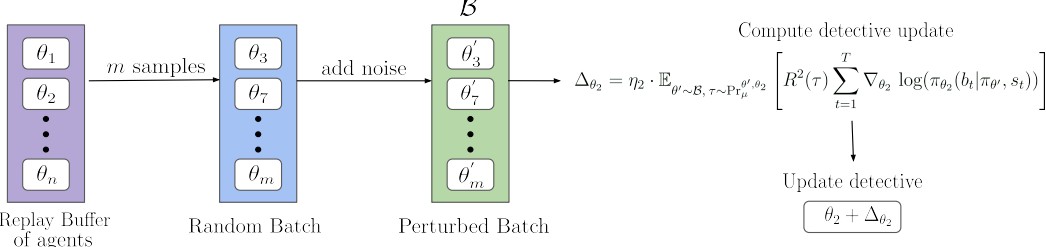

Figure 1: The detective is trained using agents sampled from a replay buffer, which contains agents encountered during training. Additional noise is incorporated to broaden the range of policies.

Policy Evaluation Networks (PVN) conditions a neural network on a policy by considering the policy's behavior on a set of learned states from the environment Harb et al. (2020). This aligns closely with our QA idea for conditioning the detective opponent on the agent's policy. However, the PVN representation is not dependent on the current state.

The concept of sequential social dilemmas is introduced in Leibo et al. (2017), as a temporally extended game with specific value constraints under the cooperate and defect policies. Due to its temporal extension and reward structure, the Coin Game is an instance of a sequential social dilemma.

Stackelberg Games Colman & Stirk (1998) revolve around a leader's initial action selection followed by a follower's subsequent move. The Bi-Level Actor-Critic(Bi-AC) Zhang et al. (2020) framework introduces an innovative approach for training both leader and follower simultaneously during the training period while maintaining independent executability, making it well-suited for addressing coordination challenges in MARL. In contrast to our setup, where the detective functions as a training harness discarded post-training, the Bi-AC varies by deploying both leader and follower jointly during test time (as the main concern is coordination between the leader and the follower). The interactions between the agent and the detective mirror the foundational Stackelberg setup, casting the agent as the leader and the detective as the follower.

Good Shepherd Balaguer et al. (2022) trains a best response to a learning agent, mirroring the best response to the best response idea. The authors offer two methods for training against this optimal response. First, by creating an expansive computational graph for the agent's optimization. Second, employing evolutionary strategies. Neither of these methods is scalable. Constructing a full optimization computational graph for every agent's optimization step is very inefficient. Moreover, evolutionary strategies require training the opponent against new data points each time. Our approach circumvents this problem by using a neural network to amortize the optimization process. PSRO Lanctot et al. (2017) unifies many MARL training frameworks like Independent RL, Iterated Best Response, and Fictitious Self-Play. PSRO-family methods iteratively extend a set of past policies, by adding the best response to a mixture of those past policies. In contrast to BRS, PSRO does not differentiate through the best response.

## 4 BEST RESPONSE SHAPING

Our Best Response Shaping (BRS) algorithm trains an agent by differentiating through an approximation to the best response opponent (as described in Section 4.1). This opponent, called the *detective*, conditions on the agent's policy via a question answering mechanism to select its actions (Section 4.2). Subsequently, we train the agent by differentiating through the detective using the RE-

INFORCE gradient estimator (Sutton et al., 1999) (Section 4.2.2). Also, to encourage cooperative behaviour, we propose Self-Play as a regularization method, encouraging the agent to explore cooperative policies. We further prove that this self-play is equivalent to self-play with reward sharing. The pseudo-code for BRS is provided in Algorithm 1.

## 4.1 BEST RESPONSE AGENT TO THE BEST RESPONSE OPPONENT

Our notation and definitions follow from Agarwal et al. (2021), we denote $\tau$ as a trajectory whose distribution, $\mathrm{Pr}_\mu^{\theta_1,\theta_2}(\tau)$, with initial state distribution $\mu$, is given by

$$\mathrm{Pr}_\mu^{\theta_1,\theta_2}(\tau) = \mu(s_0)\pi_{\theta_1}(a_0|s_0)\pi_{\theta_2}(b_0|\pi_{\theta_1},s_0)P(s_1|s_0,a_0,b_0)\cdots$$

Here $a$ denotes the action taken by the agent and $b$ the action taken by the opponent. The best response opponent is the policy that gets the highest expected return against a given agent. Formally, given $\theta_1$, the best response opponent policy $\theta_2^*$ solves for the following:

$$\theta_2^* = \arg\max_{\theta_2}\mathbb{E}_{\tau\sim\mathrm{Pr}_\mu^{\theta_1,\theta_2}}\left[R^2(\tau)\right] \tag{1}$$

Subsequently, we train the agent's policy to get the highest expected return against the best response agent. This training of the agent's policy is solving for the following:

$$\theta_1^{**} = \arg\max_{\theta_1}\mathbb{E}_{\tau\sim\mathrm{Pr}_\mu^{\theta_1,\theta_2^*}}\left[R^1(\tau)\right] \tag{2}$$

Note that this is a bi-level optimization problem. We hypothesize that the agent $\pi_{\theta_1^{**}}$ exhibits characteristics of a non-exploitable agent, as it learns retaliatory strategies in response to a defecting opponent, thereby creating incentives for a rational opponent to cooperate.

## 4.2 DETECTIVE OPPONENT TRAINING

In deep reinforcement learning, the training of agents relies on the utilization of gradient-based optimization. Consequently, we need a differentiable opponent approximating a best response opponent. We call this opponent the *detective*. The detective's policy conditions on the agent's policy in addition to the state of the environment, which we denote $\pi_{\theta_2}(a|\pi_{\theta_1},s)$. We train the detective to maximize its own return against various agents. Formally, the detective is trained by the following gradient step:

$$\nabla_{\theta_2}\mathbb{E}_{\theta_1\sim\mathcal{B}}\mathbb{E}_{\tau\sim\mathrm{Pr}_\mu^{\theta_1,\theta_2}}\left[R^2(\tau)\right] \tag{3}$$

where $\mathcal{B}$ represents a distribution of diverse policies for agent 1. It should be noted that the detective is trained online and the replay buffer, $\mathcal{B}$, is being updated with the current agent parameters.

### 4.2.1 CONDITIONING ON AGENT'S POLICY

To effectively train the agent's policy against the detective using gradient ascent on the agent's return, it is essential to establish a differentiable mechanism for the detective's conditioning. In scenarios involving toy environments with simple policy spaces, a straightforward approach of directly incorporating the agent's parameters as an input to the detective's policy works. However, it proves to be infeasible for larger policy spaces. This becomes particularly challenging when the agent's policy is represented by a neural network, as conditioning on the parameters would require an impractical number of samples. To address this limitation in more complex cases, we employ two strategies:

**State aware conditioning**. Extracting a general representation of the agent's behavior is a complex task. Instead, the detective extracts a representation for the current state of the game.

**Conditioning on behavior**. The detective queries the behaviour of the agent on various states of the game. To do so, it evaluates the agent's action probabilities (answers) on a state of the game (questions). Formally, let $\mathcal{Q}_\psi(\theta_1,s)$ be the function used by the detective to extract a state-aware representation of the agent. We call $\mathcal{Q}$ a question answering (QA) function if $\mathcal{Q}$ can be expressed as only having access to the policy function, i.e. $\mathcal{Q}_\psi(\pi_{\theta_1},s)$. There are many possible ways to architect a QA function. Next, we outline a method that has shown success in the Coin Game.

### 4.2.2 Simulation Based Question Answering

The behavior of the agent in possible continuations of the game starting from state $s$ holds valuable information. More specifically, we can assess the behavior of the agent against a random agent starting from game state $s$. Formally Let $\delta_A$ be defined as the following where $\tau$ is a trajectory starting from state $s$ at time $t$:

$$\delta_A := \mathbb{E}_{\tau \sim \Pr_\mu^{\theta_1, \theta_r}} \left[ R^r(\tau) | s_t = s \right] \tag{4}$$

where $\pi_{\theta_r}$ is an opponent that chooses action $A$ at time $t$ and afterwards samples from a uniform distribution over all possible actions:

$$\pi_{\theta_r}(a_i = A | s_i) = \begin{cases} \frac{1}{|\mathcal{A}|} & \text{if } i > t \\ \mathbb{1}_{\{a_i = A\}} & \text{if } i = t \end{cases} \tag{5}$$

Detective estimates $\delta_A$ by monte-carlo rollouts of the game to a certain length between the agent and the random opponent, $\pi_{\theta_r}$. We denote the estimate of $\delta_A$ by $\hat{\delta}_A$. Then we define $\mathcal{Q}^{\text{simulation}} = [\hat{\delta}_{A_1}, \hat{\delta}_{A_2}, \cdots, \hat{\delta}_{A_{|\mathcal{A}|}}]$. The number of samples used to estimate the returns of the game and the length of the simulated games are considered hyperparameters of $\mathcal{Q}^{\text{simulation}}$ QA. Note that the $\mathcal{Q}^{\text{simulation}}$ can be differentiated with respect to agent's policy parameters via REINFORCE (Sutton et al., 1999) term. Specifically, we use the DICE operator (Foerster et al., 2018).

### 4.2.3 Differentiating Through the Detective

The agent's policy is trained to maximize its return against the detective opponent via REINFORCE gradient estimator. However, because the detective's policy is taking the agent's policy as input, the REINFORCE term will include an additional detective-backpropagation term over the usual REINFORCE term:

$$\mathbb{E}_{\tau \sim \Pr_\mu^{\theta_1, \theta_2}} \left[ R^1(\tau) \sum_{t=1}^T \left[ \nabla_{\theta_1} \log(\pi_{\theta_1}(a_t | s_t)) + \underbrace{\nabla_{\theta_1} \log(\pi_{\theta_2}(b_t | \pi_{\theta_1}, s_t))}_{\text{detective-backpropagation term}} \right] \right] \tag{6}$$

This extra term can be thought of as the direction in policy space in which changing the agent's parameters encourages the detective to take actions that increase the agent's own return.

### 4.2.4 Cooperation Regularization via Self-Play with Reward Sharing

Agents that are trained against rational opponents tend to rely on the assumption that the opposing agent is lenient towards their non-cooperative actions. This reliance on rational behavior allows them to exploit the opponent to some extent. Consequently, they may not effectively learn to cooperate with their own selves. In scenarios where the objective is to foster more cooperative behavior, particularly encouraging the agent to cooperate with itself, a straightforward approach is to train the agent in a self-play setting, assuming that the opponent's policy mirrors the agent's policy. Formally, we update the agent using the following update rule:

$$\nabla_{\theta_1} \mathbb{E}_{\tau \sim \Pr_\mu^{\theta_1, \theta_1}} \left[ R^1(\tau) \right] \tag{7}$$

We prove that in symmetric games like IPD and Coin Game, this is equivalent to training an agent with self-play with reward sharing (see proof in §D). This training brings out the cooperative element of general-sum games. In zero-sum games, this update will have no effect as the gradient would be zero (see proof in §D). We refer to this regularization loss term as Self-Play with reward sharing throughout the paper. We also ablate BRS-NOSP where we skip the self-play loss to study its effect.

## 5 Experiments

### 5.1 Iterated Prisoner's Dilemma

Following Foerster et al. (2018), we study Iterated Prisoner's Dilemma (IPD) game where the agents observe the last actions taken by the agents. Therefore, all possible agent observations are $\mathcal{S} =$

---

**Algorithm 1** BRS pseudo code: a single iteration

---

**Input:** Replay Buffer of Agent Parameters $\mathcal{B}$, Agent parameters $\theta^1$, Detective parameters $\theta_2$, learning rates $\alpha_1, \alpha_2, \alpha_3$, Standard Error of Noise $\sigma$
**Train Detective vs. Sampled Agent:**
Sample agent parameter $\theta_1\prime$ from $\mathcal{B}$
$\theta_1\prime \leftarrow \theta_1\prime + z$, where $z \sim \mathcal{N}(0, \sigma)$
Rollout trajectory $\tau_2$ using policies $(\pi_{\theta_1\prime}, \pi_{\theta_2})$
$\theta_2 \leftarrow \theta_2 + \alpha_2 R^2(\tau_2) \sum_{t=1}^{T} \nabla_{\theta_2} \log(\pi_{\theta_2}(a_t | \pi_{\theta_1\prime}, s_t))$
**Train Agent vs. Detective:**
Rollout trajectory $\tau_1$ using policies $(\pi_{\theta_1}, \pi_{\theta_2})$
$\theta_1 \leftarrow \theta_1 + \alpha_1 R^1(\tau) \sum_{t=1}^{T} \nabla_{\theta_1} \log(\pi_{\theta_1}(a_t | s_t)) + \nabla_{\theta_1} \log(\pi_{\theta_2}(b_t | \pi_{\theta_1}, s_t))$
**Train Agent in Self Play:**
Rollout trajectory $\tau_3$ using policies $(\pi_{\theta_1}, \pi_{\theta_1})$
$\theta_1 \leftarrow \theta_1 + \alpha_3 R^1(\tau_3) \sum_{t=1}^{T} \nabla_{\theta_1} \left[ \log(\pi_{\theta_1}(a_t | s_t)) + \log(\pi_{\theta_1}(b_t | s_t)) \right]$
**Update Replay Buffer:**
Push $\theta_1$ to $\mathcal{B}$
**Output:** $\theta_1, \theta_2$

---

$\{\mathrm{C, CC, CD, DC, DD}\}$, where $C$ is the initial state, and each agent's policy can be described by the probability of cooperation for each $s \in \mathcal{S}$. We consider the IPD game that is six steps long. As shown by Foerster et al. (2018) and Zhao et al. (2022), training two naïve-learning agents leads to strategies that always defect. Although this is a Nash Equilibrium, both agents receive negative returns.

We test our method by training the agent against a tree search detective. The tree search detective constructs a tree, commencing from the current state. During this process, the agent's actions are sampled from the agent's policy, while the tree branches explore all possible choices for the detective's actions. The detective selects the actions that maximize its return, i.e. the actions that construct the best response path within the tree. The agent receives the return that corresponds to this particular path (see §F for details). Our agent is a two-layer MLP that receives the five possible states and outputs the probability of cooperation. We choose an MLP to showcase the possibility of training neural networks via BRS. We update our agent policy via policy gradient. As shown in Figure 2 the BRS agent learns tit-for-tat(TFT) policy.

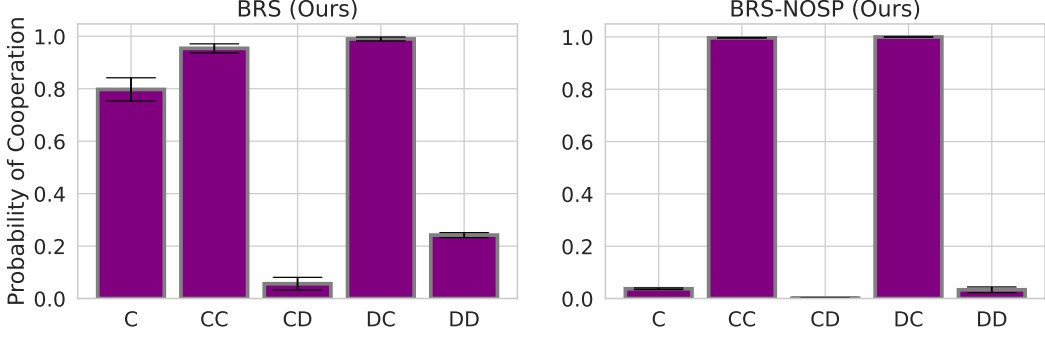

Figure 2: Illustration of the policies of agents trained with BRS and BRS-NOSP in a finite Iterated Prisoner's Dilemma game of length 6. The agents are trained against a tree search detective maximizing its own return. BRS agents learn tit-for-tat, a policy that cooperates initially and mirrors the opponent's behavior thereafter. BRS-NOSP agents learn cynic-tit-for-tat (CTFT), they defect initially but mirror the opponent's behavior thereafter.

## 5.2 THE COIN GAME

The Coin Game, introduced by Foerster et al. (2018), is a two-player general sum game that takes place in a grid. The game involves two players: the red player and the blue player. At each episode, a coin, either red or blue, spawns somewhere in the grid and players compete to pick it up. The

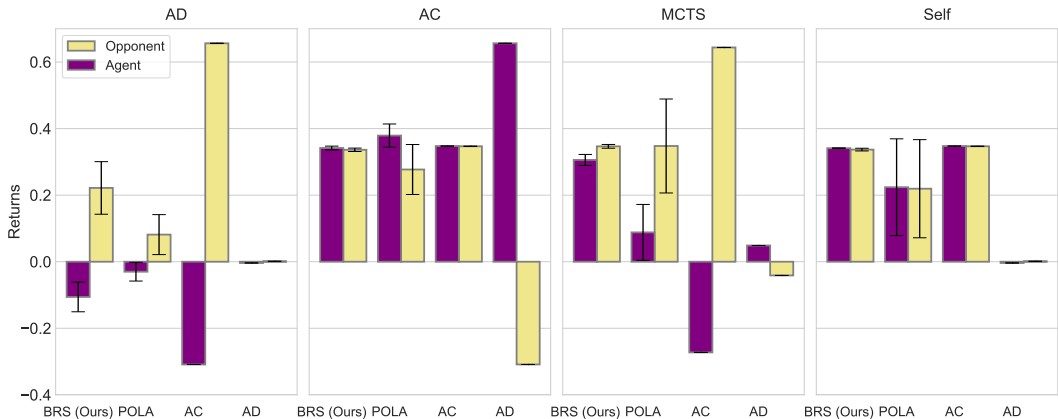

Figure 3: Comparison of BRS and POLA on Coin Game. We evaluate the agent's returns versus different opponents: Always Defect opponent (AD); Always Cooperate opponent (AC), A Monte Carlo Tree Search opponent (MCTS) and agent's performance against itself (Self).

color of the spawning coin changes after each coin is taken. If a player picks any coin, they receive a positive reward of +1. If the coin corresponding to their color is picked by the other player, they are punished with a negative reward of −2. Ideally, players should cooperate by taking only the coins of their associated color in fear of future retaliation from the other agent.

We follow Zhao et al. (2022) in training a GRU (Cho et al., 2014) agent on a $3 \times 3$ sized Coin Game with a game length of $50$ and a discount factor of $0.96$. The detective opponent is also a GRU agent with an MLP that conditions on the result of the QA (for more details see §A). We evaluate BRS and POLA agents against four policies: an opponent that always takes the shortest path towards the coin regardless of the coin's color (Always Defect), an opponent that takes the shortest path towards its associated coin but never picks up the agent's associated coin (Always Cooperate), a Monte Carlo Tree Search opponent that evaluates multiple rollouts of the game against the agent in order to take an action (MCTS), and itself (Self). Note that the MCTS will approximate the best response opponent. Figure 3 visually presents the evaluation metrics for the BRS and POLA agents. In the subsequent paragraphs, we present a comprehensive analysis and interpretation of these results.

**Does a best response opponent cooperate with the agent?** For a given environment, the opponents will learn the best response to our agent. We want those opponents to figure out that they cannot do better than Always Cooperate against. In other words, defecting against our agent would decrease their return. The MCTS approximates the best response opponent. As shown in Figure 3, the MCTS and BRS are always cooperating with each other[1]. In contrast, the MCTS does not fully cooperate with POLA. The MCTS secured a higher return than Always Cooperate against POLA via defecting.

**Does the agent retaliate against Always Defect?** If an agent never retaliates against Always Defect, its maximum return would be close to Always Cooperate against Always Defect which is -0.31, shown in Figure 5. BRS gets an average return of -0.11 against Always Defect indicating it retaliates v.s. defects. However, POLA gets -0.03 against Always Defect indicating stronger retaliation.

**Does the agent cooperate with itself?** As shown in Figure 3 BRS agents get a return of 0.33 against themselves which is very close to Always Cooperate vs Alwayas Cooperate return of 0.34. POLA agents get a retun of 0.23 against themselves indicating less cooperation. In summary, BRS agents are more suitable as a retaliatory cooperative policy. While the best response to them is always cooperation, they also fully cooperate with themselves. In contrast, the best response to POLA agents is not full cooperation, and also they do not fully cooperate with themselves.

In summary, BRS resembles TFT more closely than POLA. First, the best response to TFT is co-operation, which is true for BRS but not POLA. Second, unlike POLA, BRS agents (like TFT) cooperate well among themselves.

---

[1]Note that the return of both agents is very close to Always Cooperate vs Always Cooperate.

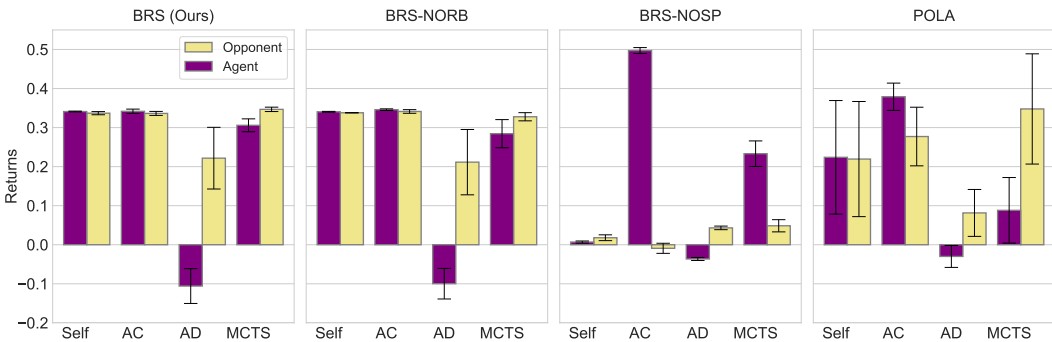

Figure 4: BRS-NORB is equivalent to BRS, with no replay buffer and no added noise. Its performance is close to BRS with more variance. BRS-NOSP is equivalent to BRS but with no self-play.

### 5.3 REPLAY BUFFER ABLATION

As shown in Algorithm 1 we train the detective against agents sampled from a replay buffer. Also, we add a small noise to the sampled agent parameters. If we had no replay buffer and we did not add any noise, would BRS still achieve the same results? In Figure 4 BRS-NORB has the same training setup as BRS with no replay buffer and no noise. While BRS-NORB has higher variance in performance than BRS, its performance is close to BRS.

### 5.4 SELF-PLAY ABLATION

As show in Algorithm 1 self-play guides the search towards cooperative policies. What policies would BRS learn if we exclude the self-play? We find that BRS-NOSP learns policies that resemble ZD-Extortion Press & Dyson (2012). They exploit opponent's rationality to increase their return and don't cooperate with themselves(see details in §E) which renders them suboptimal for scenarios where social-welfare is important.

## 6 LIMITATIONS

This paper focuses on the implementation of our proposed idea in two-player games. Extending this approach to more than two players is non-trivial[2]. Additionally, the detective agent approximates the best response opponent by training against a diverse set of agents. In this study, we introduce a replay buffer that contains previous agents encountered during training as a proxy for a diverse agent set. In 5.3 we showed BRS works even with no replay buffer on the Coin Game. Nevertheless, for more complex settings, this level of diversity may be insufficient.

## 7 CONCLUSION

Motivated by learning with learning awareness as a framework to learn reciprocity-based cooperative policies, we introduced BRS. BRS differentiates through an opponent that approximates the best response. To enable the opponent to condition on agent's policy, we introduced a novel differentiable state-aware conditioning mechanism. Additionally, self-play was incorporated to constrain the search space to self-cooperative policies. We evaluated BRS agents in detail on the Coin game. The BRS agent reaches a policy where always cooperate is the best response. We hope this work helps improving the scalability and non-exploitability of agents in Multi Agent Reinforcement Learning enabling agents that learn reciprocation-based cooperation in complex games.

---

[2]One idea to extend BRS to more than two players is to assume all the opponents as a single combined "detective" opponent. However, we have not studied the effect of such an assumption and we leave that to future work.

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

## A    EXPERIMENTAL DETAILS

### A.1    IPD

In IPD experiments, we are experimenting on IPD with 6 steps and discount factor of 1., i.e. no discount factor. The payoff matrix of the IPD game is shown in 1.

| Player 1 / Player 2 | Cooperate | Defect |
|---|---|---|
| Cooperate | $-1$ / $-1$ | $-3$ / $0$ |
| Defect | $0$ / $-3$ | $-2$ / $-2$ |

Table 1: Payoff matrix for the prisoner's dilemma game

Our agent's policy is parameterized by a two-layer MLP (Multi-Layer Perceptron) with a tanh non-linearity. The choice of tanh non-linearity is motivated by its smoothing effect and its ability to prevent large gradient updates.

During training, the agent is trained against the Tree Search Detective (TSD) (see Appendix F) using a policy gradient estimator. We employ a learning rate of $3e-4$ with the SGD (Stochastic Gradient Descent) optimizer. In the BRS experiments, the Self-Play with reward sharing loss is optimized using SGD with the same learning rate of $3e-4$. To reduce variance, the policy gradients incorporate a baseline.

For replicating the exact results presented in the paper, we provide the code in Appendix B. Running the code on an A100 GPU is expected to take approximately an hour. The plots and error bars are averaged over 10 seeds for both BRS and BRS-NOSP. The hyperparameter search was conducted by iterating over various learning rates including $(1e-4, 3e-4, 1e-3)$, and the optimizers were explored between SGD and Adam.

### A.2    COIN GAME

**The game**. Our coin game implementation exactly follows the POLA implementation Zhao et al. (2022). Similar to POLA, we also experiment with the game length of $50$ and a discount factor of $0.96$.

**Agent's architecture**. In the coin game, we have an actor-critic setup. The policy of our agent is parameterized by a GRU (Gated Recurrent Unit) architecture, following the approach outlined in the POLA repository (source). However, we introduce a modification compared to POLA by including a two-layer MLP on top of the observations before they are fed into the GRU instead of a single-layer MLP. Additionally, we utilize two linear heads to facilitate separate learning for policy and value estimation.

**Detective's architecture**. The architecture of the detective is as follows: The sequence of observations is fed into a GRU (Gated Recurrent Unit), which is the same architecture used by the agent. At each time step, the agent's representation is extracted using the QA (Question-Answering) module of the detective. In our experiments, we employed 16 samples of continuing the game for the next 4 steps from the current state. Subsequently, the output of the QA module and the GRU are concatenated and passed through a two-layer MLP with ReLU non-linearities. The resulting output from this MLP is then fed into a linear layer for estimating the value (critic), and a linear layer for determining the policy (actor).

**Separate optimizers for the two terms**. The agent uses separate optimizers for the two terms in the policy gradient. That is, it uses two separate optimizers for the two terms indicated in 8.

$$\mathbb{E}_{\tau \sim \mathrm{Pr}_\mu^{\theta_1, \theta_2}} \left[ R^1(\tau) \sum_{t=1}^{T} \left[ \underbrace{\nabla_{\theta_1} \log(\pi_{\theta_1}(a_t | s_t))}_{\text{Term 1}} + \underbrace{\nabla_{\theta_1} \log(\pi_{\theta_2}(b_t | \pi_{\theta_1}, s_t))}_{\text{Term 2}} \right] \right], \tag{8}$$

**Losses and optimizers**. The value functions in our setup are trained using the Huber loss. On the other hand, the policies are trained using the standard policy gradient loss with Generalized Advantage Estimation (GAE) (Schulman et al., 2018). However, it is important to note that our hyperparameter search led us to set the GAE parameter, $\lambda$, to 1, which results in an equivalent estimation of the advantage using the Monte-Carlo estimate. This choice is similar to the hyperparameters reported by POLA (source).

In the BRS-NOSP experiments, the agent's policy is trained using a learning rate of $1e-3$, while in the BRS experiments, an Adam optimizer with a learning rate of $3e-4$ is utilized. The value functions of both the agent and the detective in all experiments are trained using Adam with a learning rate of $3e-4$. Similarly, the detective's policy is trained using Adam with a learning rate of $3e-4$ in all experiments.

**Replay buffer of previous agents**. During the training, we keep a replay buffer of previous agents seen during the training. In BRS-NOSP experiments we keep $2048$ previous agents and in BRS experiments we keep the last $512$ agents. For training the detective, we sample a batch from this replay buffers uniformly. We add a normal noise with variance of $0.01$ to the parameters of these agents to ensure the detective is trained against a diverse set of agents.

**Hyperparameter search**. We conducted a hyperparameter search using random search over the configurations explained Table 2. the entropy coefficient $\beta$, which is multiplied by the entropy of the log probabilities associated with the actions of the corresponding player, is added to the policy gradient loss of the corresponding player for controlling the exploration-exploitation trade-off.

**Plots and error bars**. The results on the paper are computed over three seeds for the BRS, BRS-NOSP, BRS-NOSP-NORB, and BRS-NOSP-NORB and six seeds for POLA. It is worth noting that the error bars are calculate over seeds, i.e. checkpoints. The result of games between each pair of agents is averaged over 32 independent games between those two agents.

| Hyperparameter | Values |
|---|---|
| inner game length in QA | 4, 8, 12, 16 |
| samples in QA | 16, 64, 256, 1024 |
| replay buffer of agent's size | 10, 512, 4096, 16384 |
| value learning algorithm | TD-0, Monte-Carlo |
| GAE $\lambda$ | 0.9, 0.96, 0.99, 0.999, 1.0 |
| agent policy gradient learning rate | 0.001, 0.0003 |
| agent entropy $\beta$ | 0.0, 1.0, 2.0, 5.0, 10.0 |
| detective entropy $\beta$ | 0.0, 1.0, 2.0, 5.0, 10.0 |

Table 2: Hyperparameter search options

**Compute**. Our runs are run for 48 hours on a single A100 GPU with 40 Gigabytes of RAM[3].

**Batch size**. We use a batch size of 128.

**POLA agent's training**. To evaluate the POLA agents, we trained them by executing the POLA repository here (Zhao et al., 2022).

## B    REPRODUCING RESULTS

### B.1    IPD

To replicate the results on IPD (Iterated Prisoner's Dilemma), please refer to the instructions available at here. By running the provided Colab notebook, you will obtain the IPD plot that is included in the paper.

---

[3]A single A100 gpu is 80 Gigabyte, but it can be split into two equivalent 40 Gigabyte equivalents and we train on one of these splits

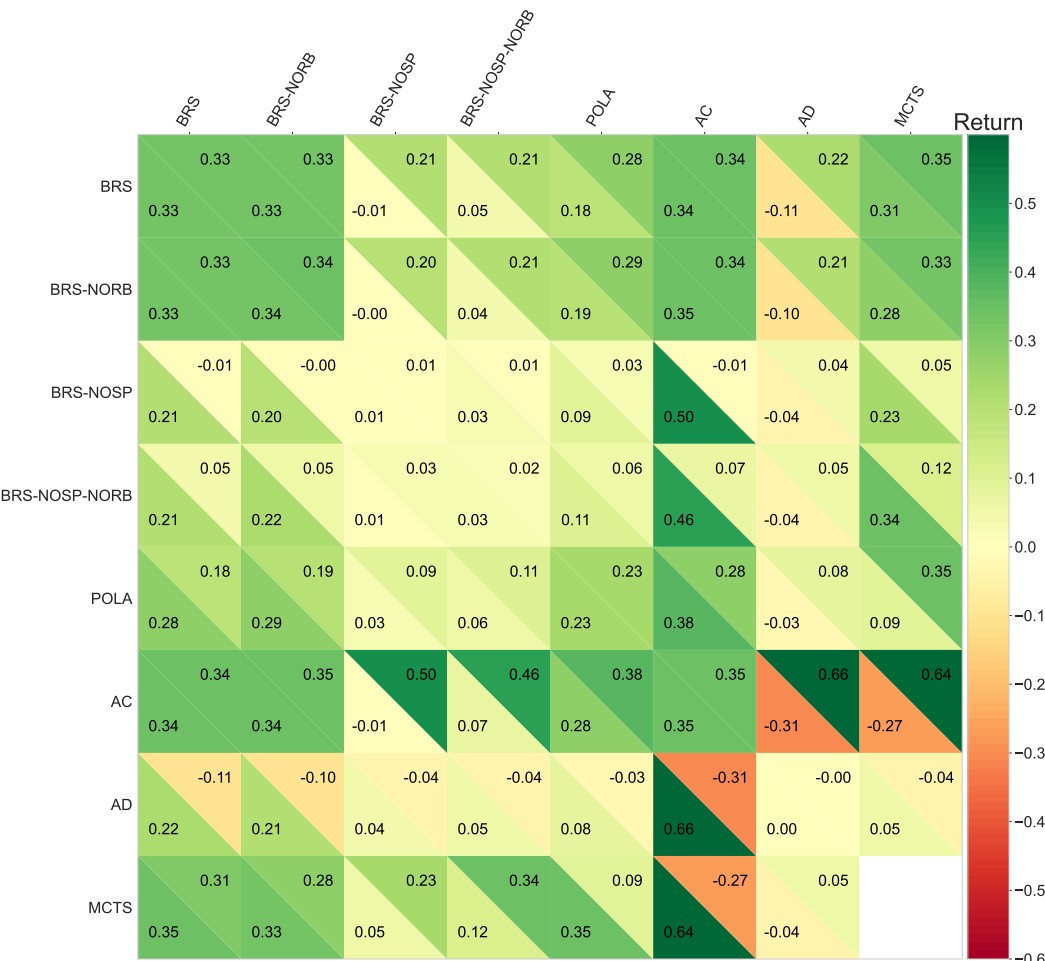

Figure 5: The presented figure illustrates the outcomes of 1-vs-1 Coin games lasting 50 rounds, involving a range of agents. The return achieved by each agent is documented within the corresponding cell. The reported returns are an average across 32 independent games. It is important to note that there are no games recorded between the MTCS agent and itself as it is not possible.

## B.2 COIN GAME

To replicate the outcomes of the coin game, please refer to the instructions available at here. In essence, the provided guidelines encompass training scripts designed for the purpose of training agent checkpoints. Subsequently, there is an exporting phase in which these checkpoints are transformed into their lightweight counterparts. Finally, a script is provided to facilitate the execution of a league involving multiple agents.

## C LEAGUE RESULTS

In order to visualize the results of our training in complete detail, in Figure 5 we visualize a matrix, in the format of a heatmap, of the returns of various agents against each other. All the results are averaged over 32 independent games between the corresponding agents. The game is the Coin game of length 50. [4]

---

[4]Note that there is no meaning to train MCTS against MCTS because the MCTS needs to roll-out the agent's policy to choose an action. However, MCTS against MCTS implies an infinite loop of rolling out the other agent's policy

## D  SELF-PLAY

**Lemma D.1.** *Denote* $o \in \mathcal{S}$ *to be the state* $s \in \mathcal{S}$ *from the perspective of the opponent. For a symmetric game, if it holds that* $\mu(s_0) = \mu(o_0)$ *for all* $s_0, o_0 \in \mathcal{S}$*, then*

$$\mathbb{E}_{\tau \sim \mathrm{Pr}_\mu^{\theta_1,\theta_1}} \left[ R^1(\tau) \right] = \mathbb{E}_{\tau \sim \mathrm{Pr}_\mu^{\theta_1,\theta_1}} \left[ R^2(\tau) \right]$$

*where* $R^2 := \sum_{t=0}^\infty \gamma^t r^2(o_t, b_t, a_t)$ *and* $r^2$ *denotes* $r^1$ *from the perspective of the opponent.*

*Proof.* Denote $\bar{\tau} = o_0, b_0, a_0, o_1, \cdots$, then notice that

$$\mu(s_0)\pi_{\theta_1}^1(a_0|s_0)\pi_{\theta_1}^1(b_0|o_0)P(s_1|s_0,a_0,b_0)\cdots = \mu(o_0)\pi_{\theta_1}^1(b_0|o_0)\pi_{\theta_1}^1(a_0|s_0)P(o_1|o_0,b_0,a_0)\cdots$$
$$\iff \mathrm{Pr}_\mu^{\theta_1,\theta_1}(\tau) = \mathrm{Pr}_\mu^{\theta_1,\theta_1}(\bar{\tau})$$

now by symmetry we have that $r^1(s_t, a_t, b_t) = r^2(o_t, b_t, a_t)$, therefore

$$\mathbb{E}_{\tau \sim \mathrm{Pr}_\mu^{\theta_1,\theta_1}} \left[ R^1(\tau) \right] = \mathbb{E}_{\tau \sim \mathrm{Pr}_\mu^{\theta_1,\theta_1}} \left[ \sum_{t=0}^\infty \gamma^t r^1(s_t, a_t, b_t) \right]$$
$$= \sum_\tau \mathrm{Pr}_\mu^{\theta_1,\theta_1}(\tau) \sum_{t=0}^\infty \gamma^t r^1(s_t, a_t, b_t)$$
$$= \sum_{\bar{\tau}} \mathrm{Pr}_\mu^{\theta_1,\theta_1}(\bar{\tau}) \sum_{t=0}^\infty \gamma^t r^2(o_t, b_t, a_t)$$
$$= \mathbb{E}_{\tau \sim \mathrm{Pr}_\mu^{\theta_1,\theta_1}} \left[ R^2(\tau) \right]$$

where we just rename $\bar{\tau}$ in the last equality. ∎

Proposition D.2 states that the gradient in Equation 7 is equivalent to that of self-play with reward-sharing.

**Proposition D.2.** *For a symmetric game,*

$$\nabla_{\theta_1} \mathbb{E}_{\tau \sim \mathrm{Pr}_\mu^{\theta_1,\theta_1}} \left[ R^1(\tau) \right] \propto \left[ \nabla_{\theta_1} \mathbb{E}_{\tau \sim \mathrm{Pr}_\mu^{\theta_1,\theta_2}} \left[ R^1(\tau) + R^2(\tau) \right] + \nabla_{\theta_2} \mathbb{E}_{\tau \sim \mathrm{Pr}_\mu^{\theta_1,\theta_2}} \left[ R^1(\tau) + R^2(\tau) \right] \right]_{\theta_2 = \theta_1}.$$

*Proof.* We write the gradient as follows:

$$\nabla_{\theta_1} \mathbb{E}_{\tau \sim \mathrm{Pr}_\mu^{\theta_1,\theta_1}} \left[ R^1(\tau) \right] = \sum_\tau R^1(\tau)\nabla_{\theta_1}\mathrm{Pr}_\mu^{\theta_1,\theta_1}(\tau)$$
$$= \sum_\tau R^1(\tau)\mathrm{Pr}_\mu^{\theta_1,\theta_1}(\tau)\nabla_{\theta_1}\log\mathrm{Pr}_\mu^{\theta_1,\theta_1}(\tau)$$
$$= \sum_\tau R^1(\tau)\mathrm{Pr}_\mu^{\theta_1,\theta_1}(\tau)\nabla_{\theta_1}\log\mu(p_0)\pi_{\theta_1}^1(a_0|s_0)\pi_{\theta_1}^1(b_0|o_0)\cdots$$
$$= \sum_\tau R^1(\tau)\mathrm{Pr}_\mu^{\theta_1,\theta_1}(\tau)\sum_{t=0}^\infty \nabla_{\theta_1}\log\pi_{\theta_1}^1(a_t|s_t) + \nabla_{\theta_1}\log\pi_{\theta_1}^1(b_t|o_t)$$
$$= \mathbb{E}_{\tau \sim \mathrm{Pr}_\mu^{\theta_1,\theta_1}} \left[ R^1(\tau)\sum_{t=0}^\infty \nabla_{\theta_1}\log\pi_{\theta_1}^1(a_t|s_t) + \nabla_{\theta_1}\log\pi_{\theta_1}^1(b_t|o_t) \right].$$

Now by symmetry and Lemma D.1. we have

$$\mathbb{E}_{\tau \sim \mathrm{Pr}_\mu^{\theta_1,\theta_1}} \left[ R^1(\tau) \right] = \mathbb{E}_{\tau \sim \mathrm{Pr}_\mu^{\theta_1,\theta_1}} \left[ R^2(\tau) \right],$$

and by linearity of expectation,

$$\mathbb{E}_{\tau \sim \mathrm{Pr}_\mu^{\theta_1,\theta_1}} \left[ R^1(\tau) \right] \propto \mathbb{E}_{\tau \sim \mathrm{Pr}_\mu^{\theta_1,\theta_1}} \left[ R^1(\tau) + R^2(\tau) \right].$$

Hence

$$
\begin{aligned}
\nabla_{\theta_1} \mathop{\mathbb{E}}_{\tau \sim \mathrm{Pr}_\mu^{\theta_1,\theta_1}} \left[ R^1(\tau) \right] \quad &\propto \quad \mathop{\mathbb{E}}_{\tau \sim \mathrm{Pr}_\mu^{\theta_1,\theta_1}} \left[ \left( R^1(\tau) + R^2(\tau) \right) \sum_{t=0}^{\infty} \nabla_{\theta_1} \log \pi_{\theta_1}^1(a_t|s_t) + \nabla_{\theta_1} \log \pi_{\theta_1}^1(b_t|o_t) \right] \\
&= \quad \left[ \mathop{\mathbb{E}}_{\tau \sim \mathrm{Pr}_\mu^{\theta_1,\theta_2}} \left[ \left( R^1(\tau) + R^2(\tau) \right) \sum_{t=0}^{\infty} \nabla_{\theta_1} \log \pi_{\theta_1}^1(a_t|s_t) + \nabla_{\theta_2} \log \pi_{\theta_2}^2(b_t|o_t) \right] \right]_{\theta_2 = \theta_1} \\
&= \quad \left[ \mathop{\mathbb{E}}_{\tau \sim \mathrm{Pr}_\mu^{\theta_1,\theta_2}} \left[ \left( R^1(\tau) + R^2(\tau) \right) \left( \nabla_{\theta_1} \log \mathrm{Pr}_\mu^{\theta_1,\theta_2}(\tau) + \nabla_{\theta_2} \log \mathrm{Pr}_\mu^{\theta_1,\theta_2}(\tau) \right) \right] \right]_{\theta_2 = \theta_1} \\
&= \quad \left[ \nabla_{\theta_1} \mathop{\mathbb{E}}_{\tau \sim \mathrm{Pr}_\mu^{\theta_1,\theta_2}} \left[ R^1(\tau) + R^2(\tau) \right] + \nabla_{\theta_2} \mathop{\mathbb{E}}_{\tau \sim \mathrm{Pr}_\mu^{\theta_1,\theta_2}} \left[ R^1(\tau) + R^2(\tau) \right] \right]_{\theta_2 = \theta_1},
\end{aligned}
$$

which was to be shown.  ∎

**Corollary D.3.** *For a symmetric, zero-sum game it holds that*

$$
\nabla_{\theta_1} \mathop{\mathbb{E}}_{\tau \sim \mathrm{Pr}_\mu^{\theta_1,\theta_1}} \left[ R^1(\tau) \right] = 0
$$

*Proof.* By definition of zero-sum game, we have that

$$
r^1(s_t, a_t, b_t) + r^2(s_t, b_t, a_t) = 0
$$

$$
\implies \sum_{t=0}^{\infty} \gamma^t \left( r^1(s_t, a_t, b_t) + r^2(s_t, b_t, a_t) \right) = 0
$$

$$
\iff R^1(\tau) = -R^2(\tau) \text{ for all } \tau
$$

From proposition D.2. we get

$$
\begin{aligned}
\nabla_{\theta_1} \mathop{\mathbb{E}}_{\tau \sim \mathrm{Pr}_\mu^{\theta_1,\theta_1}} \left[ R^1(\tau) \right] \quad &\propto \quad \left[ \nabla_{\theta_1} \mathop{\mathbb{E}}_{\tau \sim \mathrm{Pr}_\mu^{\theta_1,\theta_2}} \underbrace{\left[ R^1(\tau) + R^2(\tau) \right]}_{=0} + \nabla_{\theta_2} \mathop{\mathbb{E}}_{\tau \sim \mathrm{Pr}_\mu^{\theta_1,\theta_2}} \underbrace{\left[ R^1(\tau) + R^2(\tau) \right]}_{=0} \right]_{\theta_2 = \theta_1} \\
&= \quad \left[ \nabla_{\theta_1} 0 + \nabla_{\theta_2} 0 \right]_{\theta_2 = \theta_1} \\
&= \quad 0,
\end{aligned}
$$

completing the proof.  ∎

## E  SELF-PLAY ABLATION

### E.0.1  IPD

In IPD, as shown in Figure 2 the BRS-NOSP agents learn a variant of tit-for-tat that defects initially but has the same probability of cooperation as tit-for-tat in $\{CC, CD, DC, DD\}$. We name this policy cynic-tit-for-tat (CTFT). The best response to a cynic-tit-for-tat in an infinite IPD game is always cooperating because if the opponent defects initially, the agent will defect in the next turn. Also, CTFT does not cooperate with itself.

Furthermore, if we use the analytical differentiable returns in IPD, BRS-NOSP learns a ZD-extortion policy Press & Dyson (2012) similar to Lu et al. (2022) as shown in Figure 8. ZD-Extortion policy gains advantage by defecting to the extent that best response of the opponent is still cooperation.

### E.0.2  COIN GAME

In the Coin Game, as shown in Figure 4, the BRS-NOSP agents get a high return against the MCTS. However, the MCTS opponent gets considerably less return against BRS-NOSP than against BRS. This indicates BRS exploited the MCTS's rationality. While MCTS does better than Always Defect against the BRS-NOSP, it trades a high amount of cooperation to elicit a slight cooperation from the BRS-NOSP. In other words, teh BRS-NOSP exploites the rationality of the MCTS. Also, BRS-NOSP agents do not cooperate with themselves and they exploit Always Cooperate.

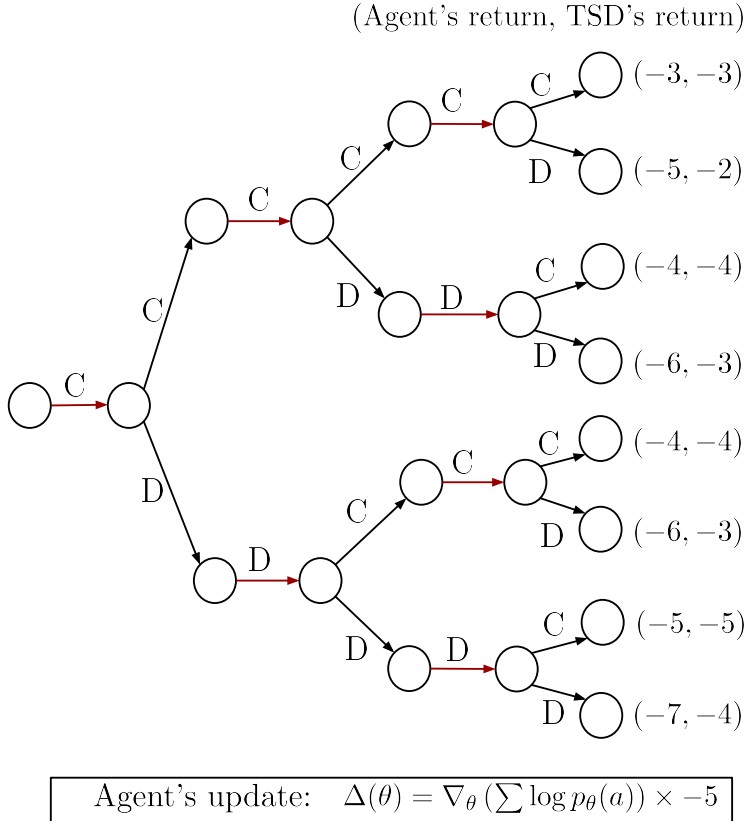

$$\text{Agent's update:} \quad \Delta(\theta) = \nabla_\theta \left( \sum \log p_\theta(a) \right) \times -5$$

Figure 6: This figure illustrates the training of the IPD agent against the TSD. TSD samples from the agent's policy, represented by red arrows in the plot, while exploring all possible actions when considering its own actions, represented by black arrows in the plot. The agent treats the TSD as a black-box algorithm and differentiates through it via REINFORCE. Note that the summation is over all log probabilities and not only over the log probabilities presnet in the path.

## F  TREE SEARCH DETECTIVE

In this section, we describe the Tree Search Detective (TSD) used in the IPD experiments.The intuition behind TSD is that by simulating all possible trajectories based on the agent's policy, the opponent can select the path that maximizes its own returns. Consequently, the agent achieves the return associated with that specific path.

TSD implements this idea. TSD builds a tree structure in which the agent's actions are directly sampled from its policy. When it comes to TSD's action, a branch is formed for each action to explore the potential outcomes of that specific action.

The agent will treat TSD as a black-box algorithm that queries the agent's policy on a set of states and returns a single return, i.e. the return that corresponds to the agent's return in the path that yielded the highest return for the TSD. This black-box can be differentiated through via policy gradient estimators. It is worth noting that when calculating the policy gradient loss, the sum of all log probabilities should be considered, not just the ones present in the chosen path. This is crucial because the agent's actions in states outside of the selected path are significant in TSD's decision-making process for selecting that particular path. This idea has been depicted in Figure 6.

## G  DETAILED RESULTS OF GAMES BETWEEN AGENTS

In Figure 7 we visualized the average result of 32 games between different agents. Note that for BRS agents we used three seeds per agent type and for POLA we used six seeds. Indeed, the POLA

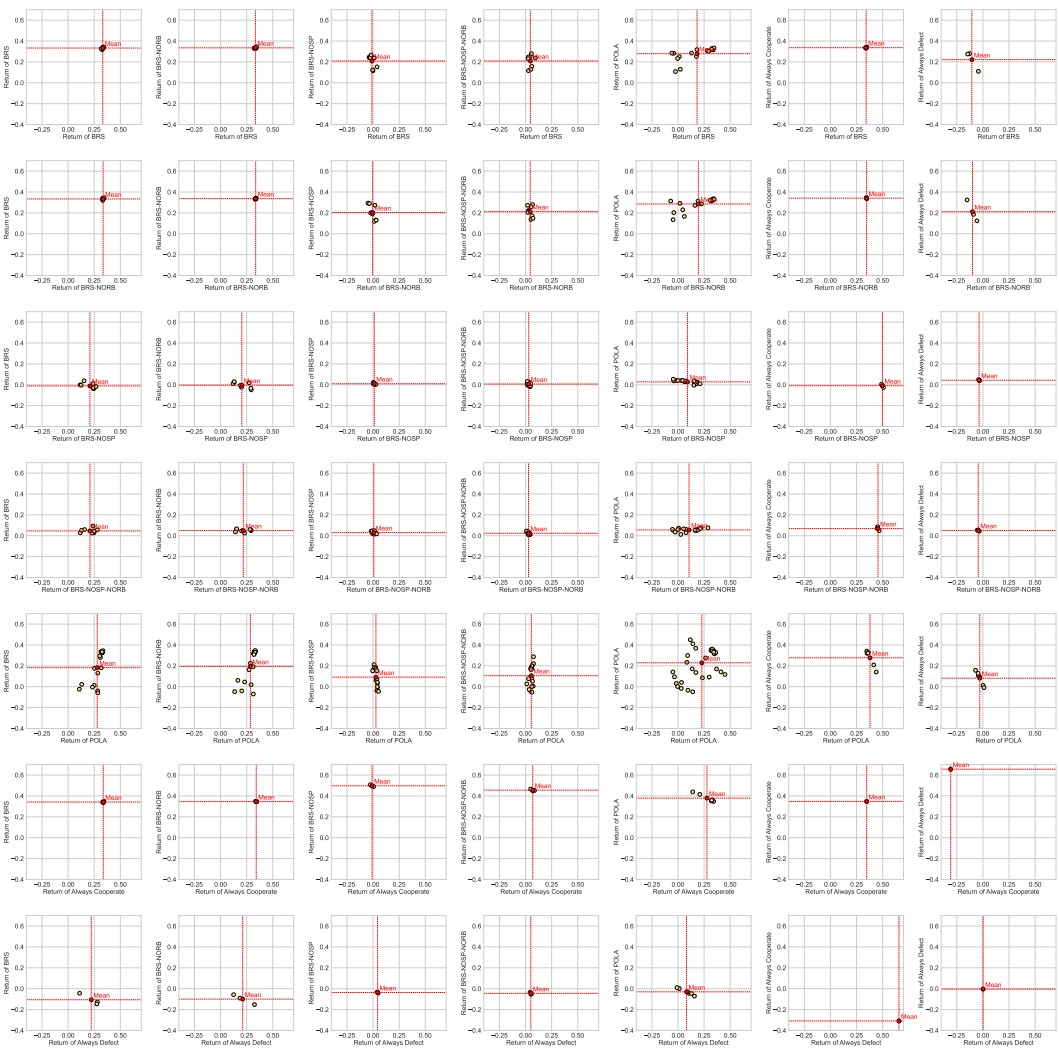

Figure 7: This figure illustrates the performance of all the agents (Including Always Cooperate and Always Defect) against each other.

agents have more variance in their performance therefore we used more seeds to compute the error bars for them.

## H   ZD-EXTORTION

Figure 8 shows that BRS without self-play learns a ZD-extortion policy as expected.

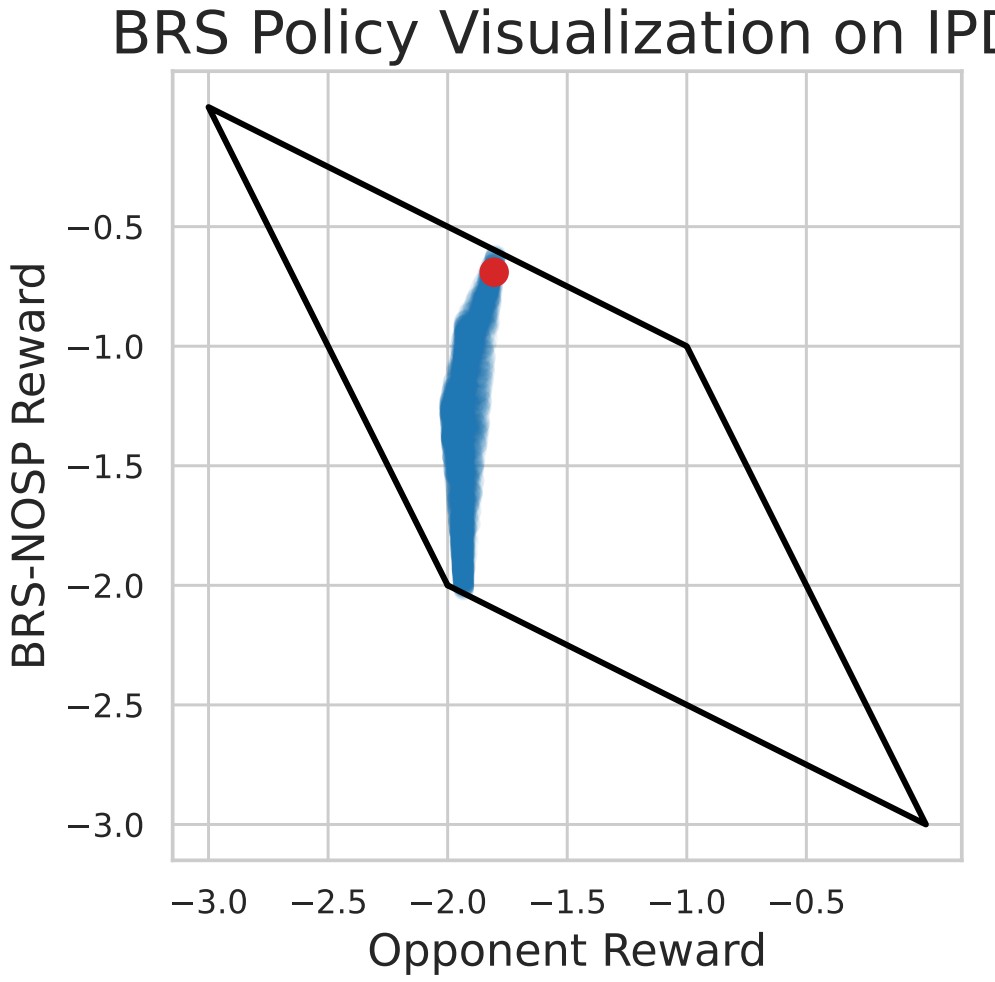

Figure 8: Visualization of BRS-NOSP's policy. Similar to Lu et al. (2022) our agent when trained to find the best response to the best response discovers a ZD-extortion policy.

## I   TRAINING CURVES OF BRS AND BRS-NOSP

Figure 9 shows the training curves of BRS and BRS-NOSP seeds.

## J   BRS VS POLA: HEAD TO HEAD RESULTS

In this section, we delve into the details of POLA vs. BRS. We sampled 32 trajectories between each POLA seed and each BRS seeds. In summary, we observe: 1) POLA seeds have higher variance in

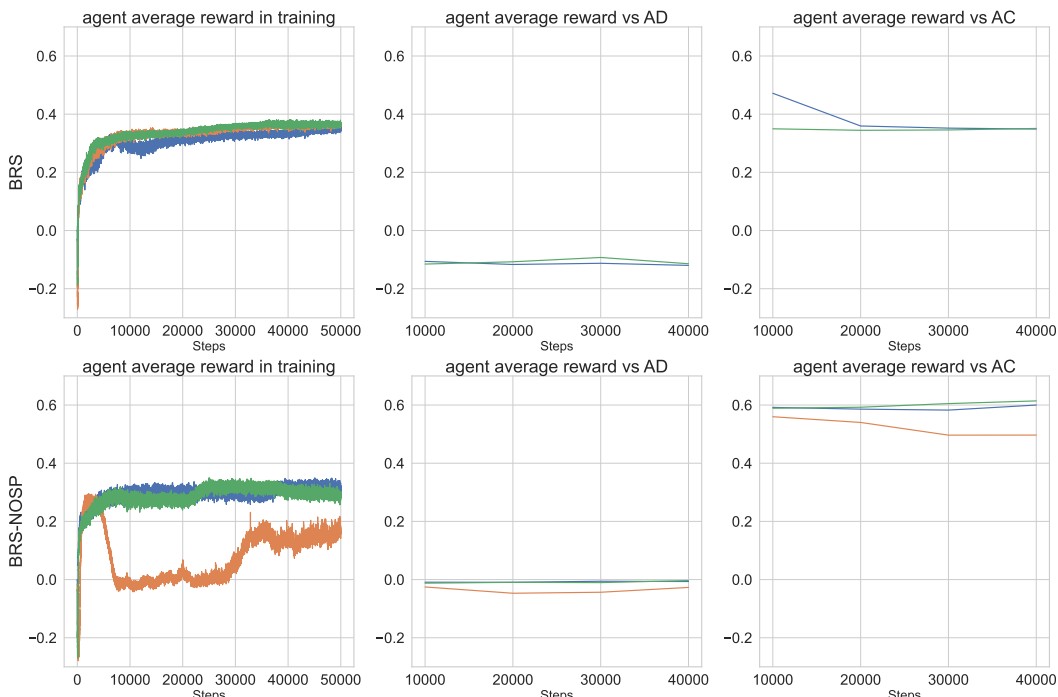

Figure 9: Training curves of BRS and BRS-NOSP during training and their evaluation against Always Defect(AD) and Always Cooperate(AC) opponent

behaviour. 2) POLA seeds break the cooperation loop much more often than BRS agents. 3) POLA agents retaliate weakly when BRS breaks the cooperation by defecting. In overall, that indicates that BRS agents are more suitable than POLA agents as reciprocation-based cooperative agents.

## J.1 RECIPROCATION-BASED COOPERATION COMPARISON

| Agent | Start | Opponent Cooperates | Opponent Defects | Opponent Defects while Agent Cooperated |
|-------|-------|---------------------|------------------|------------------------------------------|
| POLA | 0.5614 | 0.8705 | 0.1350 | 0.6944 |
| BRS | 0.9957 | 0.9894 | 0.2599 | 0.1600 |

Table 3: This table indicates the empirically estimated probability that each agent cooperates after a specific condition is met. For example, POLA cooperated with $0.6944$ probability in trajectories in which BRS defected while POLA's last action was cooperation.

We now consider empirical statistics of the observed trajectories between POLA and BRS agents in the Coin Game. Here we define cooperation as a turn in which the opponent does not take the agent's coin (and vice versa for the agent). We define for both opponent and agent defection as a turn in which they take the other's coin.

A shown in Table 3 in contrast to BRS which almost always starts with cooperation, POLA starts cooperation $0.56$ of times deviating from a TFT policy. Both POLA and BRS cooperate with high probability in case of observing that the opponent cooperated. However, BRS's policy cooperates with higher probability. Both POLA and BRS cooperate with little probability after they observe the opponent defected. While POLA cooperates with less probability than BRS which seems desirable, it should be noted that POLA seeds defect more compared to BRS seeds in general. The next column sheds lights on this. A cooperation reciprocation-based policy should defect after its cooperation is faced with opponent defection. POLA will cooperate $0.70$ times in those situations indicating lack of strong retaliation. BRS seeds cooperate $0.16$ times indicating strong retaliation. Note that these are conditional probabilities. As shown in Table 4 in these 32 trajectories we observe only 72 situations

in which POLA cooperated first and BRS defected. In 22 out of those POLA defected next and the other 50 POLA cooperated. This is a sign of weak retaliation. In contrast, we observe 950 situations in which BRS cooperated and POLA defected. In 798 out of those, BRS defected next indicating strong retaliation. In summary, these results show that POLA agents are inclined towards defecting and also they weakly retaliate while BRS agents show strong inclination towards cooperation while showing strong signs of retaliation when the opponent defects.

Table 4: Retaliation Behaviors of POLA and BRS Agents

| POLA Cooperates & BRS Defects: 72 times | | BRS Cooperates & POLA Defects: 950 times | |
|---|---|---|---|
| **POLA Defects Next** | **POLA Cooperates Next** | **BRS Defects Next** | **BRS Cooperates Next** |
| 22 times, 0.31 probability | 50 times, 0.69 probability | 798 times, 0.84 probability | 152 times, 0.16 probability |

Table 5: Shows the empirical frequency of various retaliations behaviours of POLA and BRS seeds in 32 rollouts of length 50 in the Coin Game. In 72 times BRS defected after POLA cooperated. Only 22 times out of those, POLA retaliates. In 950 times POLA defected after BRS cooperated. BRS retaliated on 798 of those.

## J.2 LEAGUE RESULTS AND ANALYSIS

Figure 10 shows the head to head results of BRS and POLA seeds. We observe that while BRS agents robustly cooperate with themselves, MCTS, and Always Cooperate the behaviour of POLA agents varies. We observe two main patterns in POLA seeds. POLA-3 and POLA-4 are exploitative, exploiting other POLA seeds and Always Cooperate. But, they cannot cooperate with themselves. While they are not exploited by MCTS in the sense of getting lower return than MCTS, their return against MCTS indicates non-cooperative rollouts. POLA-1, POLA-2, POLA-5, and POLA-6 are more cooperative - even cooperating with themselves - at the expense of being exploited by other POLA seeds and MCTS. It should be noted that for all POLA seeds the best response, approximated by the MCTS agent, is never to always cooperate. This is in contrast with BRS which not only always cooperates with itself, but also convinces the MCTS agent to always cooperate with them.

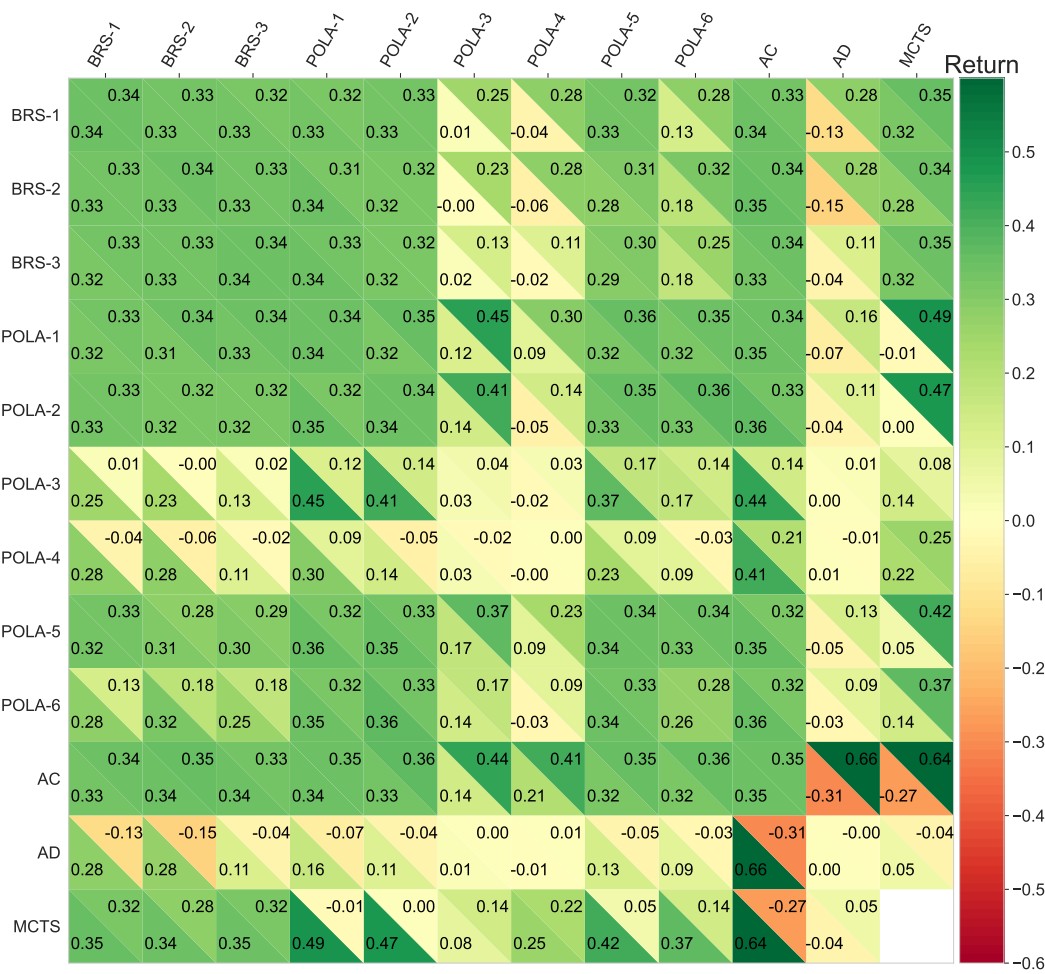

Figure 10: Head to head results of BRS and POLA seeds. Each entry is averaged over 32 independent rollouts.

