# OpenReview forum: "Best Response Shaping"
_ICLR.cc/2024/Conference — Submitted to ICLR 2024_

### Official Review · Reviewer_1vtA · 2023-10-18

**Soundness:** 2 fair
**Presentation:** 2 fair
**Contribution:** 2 fair
**Rating:** 3
**Confidence:** 5

**Summary:**

This paper proposes a learning method to discover an NE solution that discovers maximum social welfare. The idea extends the LOLA and POLA works by directly applying MARL with backpropagation through the entire opponent policy. The experiments are conducted on the IPD game and the coin game.

**Strengths:**

In general, I appreciate the attempt to extend the existing works further, from a few RL steps to the policy change via policy conditioning. However, the paper has significant technique issues.

**Weaknesses:**

### Fundamental Issue

The fundamental issue of this work is that the paper directly assumes **symmetric NEs** throughout the entire paper. If we know **in advance** that the desire NE **must** be symmetric, then all the technical derivations in this paper are correct. However, although IPD falls into this category, this remains an extremely strong assumption and you cannot really leverage this assumption during algorithmic design, which, to some extent, is a cheating strategy.

I want to remark that LOLA does not adopt parameter sharing during learning.

I also want to mention that a symmetric game does not mean that the NE is symmetric. Let's consider a particular symmetric 2-player matrix game with a payoff matrix as
```
(0, 0), (1,1)
(1,1), (0,0)
```
This is an XOR game where the NE should be two agents taking different actions, which cannot represented by policies with shared parameters. You may refer to [this paper](https://arxiv.org/abs/2206.07505) for a discussion on the XOR game.

Another example would be a modified chicken game as shown below.
```
(0, 0), (0,1)
(1,0), (-1,-1)
```
The NE solution is the same as the XOR game.

I also want to point out that the claims in Section 4.3.2 are particularly problematic.
1. "_this is equivalent to training an agent with self-play with reward sharing_". This claim is **wrong**. Equation (7) is only equivalent to **reward sharing _with parameter sharing_**. "Self-play with reward sharing'' should refer to the case where "you learn two different policies with different parameters while the game reward is shared across agents". So, self-play with reward sharing should be able to learn an NE in the XOR game while equation 7 can never due to parameter sharing
2. "_In zero-sum games, this update will have no effect as the gradient would be zero_". To be frank, I was deeply shocked by such a conclusion at the very beginning when I read this sentence. Then I find it correct but meaningless: in a symmetric and zero-sum game, the expected reward is always 0 when the players use identical policies, which means you have zero gradient everywhere.

Finally, the concept of _symmetric game_ only applies to two-player games. For general $N$-player games, you are essentially assuming the game is cooperative.

### Minor Issues

1. Section 2.1 presents a $N$-player game formulation. However, the entire paper adopts a two-player setting.
2. At the end of section 4.1, it is stated that "_Note that ($\theta^{∗∗1}$, $\theta^{∗2}$) is a nash equilibrium by definition._" This sounds wrong to me. An NE should be defined over a minimax operator rather than two separate equations. A simple counter-example here is the rock-paper-scissor game. Suppose initially you have $\theta^1$ to be rock. Then $\theta^{*2}$ will be paper, and $\theta^{∗∗1}$ becomes scissor, which is clearly not an NE.
3. The presentation of Section 4.2 is hard to follow. A few examples. What do you mean by "detective’s conditioning"? I would be better to have a formulation of what you want to present. It is also mentioned "the behavior of the agent against a random agent" in Section 4.2.2. Here the word "agent" is mentioned twice. So which is agent 1 and which is agent 2? I guess you are using agent to refer to agent 1 while the detective to agent 2 but it is not always consistent.
4. There are also notation issues. I'm always confused by which agent an action refers to. Moreover, in Equation 6, $b$ is used to denote the action from agent 2. However, the notation of $b$ is never introduced before.

**Questions:**

First of all, I think the paper should be substantially rewritten.

Besides, I would be also curious about how your algorithm will perform on cases where (1) the NE is asymmetric and (2) the parameter sharing is turned off, i.e., two agents have two separate policies.

---

> ### Author Response · Authors · 2023-11-12
>
> We thank the reviewer for their time and feedback.
>
> **Weaknesses**:
>
> **Fundamental Issue**:
>
> We would like to remind the reviewer that we are not making the assumption that the solutions to our games are symmetric Nash Equilibria: the detective and agent do not share parameters with each other therefore the claim that we assume symmetric NE’s is **false** and not incorporated into the design of the algorithm. We recognize that the main variation of the algorithm incorporates a self-play loss that, if this were to be the only loss, would make it impossible to learn non-symmetric NE’s in 1-step Matrix form games. But we would like to remind the reviewer that our scope is iterated social dilemmas.
>
> The reviewer states *“This is an XOR game where the NE should be two agents taking different actions, which cannot represented by policies with shared parameters.”*
> We present a **counter example** in the one-step history version of the XOR game. Consider the policy that randomly samples an action uniformly at random until the opponent chooses a different action, and subsequently sticks to their action thereafter. In a self-play setting (therefore a parameter sharing setting) this policy is a Nash equilibrium of the game, and corresponds to the same policy issuing different actions.
>
> Regarding the **problematic claims**:
>
> The reviewer states *“This claim is wrong. Equation (7) is only equivalent to reward sharing with parameter sharing.”* We would like to remind the reviewer that, by definition, self-play amounts to parameter sharing, therefore the original claim is **not wrong**. The reviewer also writes “Self-play with reward sharing'' should refer to the case where "you learn two different policies with different parameters while the game reward is shared across agents". This by definition cannot be self-play.
>
> The reviewer writes *“a game is both symmetric and zero-sum if and only if it is a zero game”*, this claim is **wrong**. We are only aware of the definition of “zero-game” in the context of combinatorial games (from wikipedia): “the zero game is the game where neither player has any legal options”. A fundamental assumption of combinatorial games is a turn-taking nature. Therefore any non turn-taking symmetric zero sum game (e.g. rock paper scissors) is not a zero game.
>
>
> The reviewer states “the concept of symmetric game only applies to two-player games.”, this statement is **wrong**. N-player symmetric games exist and are clearly defined. Tragedy of the Commons is one of them.
>
> Regarding **minor issues**:
>
> 1. The n-player formulation is standard in the literature and provides a general framework in which the two-player formulation is an instance.
>
> 2. This is a correct observation and we currently suspect our statement is not valid. We will remove this statement from the paper.
>
> 3. We are aware of some notational inconsistencies and we thank the reviewer for pointing them out, we will definitely correct these.
>
> 4. We agree with the reviewer that this part of the notation should be made explicit and we will incorporate it when we introduce it in section 4.1.
>
> We ask the reviewer to reconsider their score.

---

> ### Author Response · Authors · 2023-11-19
> **Discussion Period is Nearing its End**
>
> We thank the reviewer for their time and feedback.
> We hope we have addressed the raised concerns. As the discussion period is drawing to a close, we kindly ask the reviewer if there are any further concerns or points.

---

> > ### Author Response · Authors · 2023-11-22
> >
> > We thank the reviewer for their time and feedback. While we hold a strong conviction about the growing importance of our subfield in the upcoming years, we recognize it is still somewhat specialized. This requires thorough discussions to align our thought processes and offers us crucial insights to enhance the clarity of our paper. We are eager to know if the reviewer's concerns have been satisfactorily addressed.
> >
> > To summarize, we explained why our method learns non-symmetric Nash Equilibria (NEs).
> > We have demonstrated that symmetric zero-sum games do not equate to zero-games. Regarding combinatorial game theory's concept of a zero-game, where both players have no remaining legal moves, our considered games ensure that each player always has available moves. Notably, the traditional zero-game definition applies to turn-based games, whereas in our study, players act simultaneously. If the reviewer's reference to a zero-game implies a scenario where rewards are consistently zero, we wish to clarify that "symmetry" simply indicates indifference to player order—there is no advantage for player 1 over player 2, player 3, and so forth. It's important to note that symmetric multi-agent zero-sum games are abundant and are not confined to two-player scenarios, nor do they inherently restrict rewards to zero.
> >
> > Furthermore, we request the reviewer’s attention to the improvements in the updated paper, where we have corrected typographical errors, expanded the literature review, and provided a comprehensive analysis comparing POLA and BRS agents, highlighting the significant benefits of BRS over POLA.
> >
> > As the discussion period nears its conclusion, we are keen to know if we have successfully addressed the reviewer's concerns and if they are willing to reconsider their score.

---

> ### Comment · Reviewer_1vtA · 2023-11-23
>
> The authors assume the solution is symmetric with parameter sharing in the self-play process. This assumption raises two primary issues:
>
> 1. Solving the game with parameter sharing does not necessarily lead to NEs since symmetric NEs may not exist. This has been shown in the XOR game example and the chicken game example. Instead of offering examples, it would be more appropriate for the authors to rigorously prove that the solution is indeed a NE when the game is solved with parameter sharing. However, this seems unattainable, as demonstrated in the XOR game and the chicken game examples
>
> 2. Furthermore, I also want to point out that solving the game with parameter sharing and reward sharing does not necessarily lead to cooperative behavior. Consider a symmetric matrix game as shown below.
>
> ```
> (0,0), (0,1)
> (1,0), (0,0)
> ```
>
> Solving the game with reward sharing and parameter sharing by equation 7. yields a policy $\pi=(0.5,0.5)$. For $\pi=(0.5,0.5)$, the expected shared reward is $0.5\times0.5+0.5\times0.5=0.5$. However, the solution that maximizes the shared reward in this game is not symmetric, $\pi_1=(1,0)$ and $\pi_2=(0,1)$. The shared reward of this non-symmetric solution is $1$.
>
>
> Regarding the problematic claims in Section 4.3.2.
>
> 1. *"In zero-sum games, this update will have no effect as the gradient would be zero."* This is claim is correct but meaningless because in symmetric zero-sum two-player games, the expected return would always be zero if both players are using the same policy.
>
> 2. *"this is equivalent to training an agent with self-play with reward sharing".*  Self-play is not equivalent to either parameter-sharing or a symmetric policy. The error lies in the assumption that both players use precisely the same policy in self-play; in self-play, players can employ different policies at the same state.

---

### Official Review · Reviewer_FRpg · 2023-10-31

**Soundness:** 2 fair
**Presentation:** 3 good
**Contribution:** 2 fair
**Rating:** 5
**Confidence:** 3

**Summary:**

This paper proposes a novel opponent-aware framework to improve cooperation. The framework involves training a detective to approximate the opponent, training the agent based on the interaction between the detective and agent, and conducting self-play to learn to cooperate. The experiments demonstrate that the proposed method can improve social welfare.

**Strengths:**

1) The proposed method that involves training a detective to help training the agent is interesting.

**Weaknesses:**

1) My major concern is that the explanation why BRS outperforms POLA is not very clear. The authors mention that "When the opponent is specifically trained to maximize its own return against a fixed policy trained by POLA, the first exploits the former." However, it is not clear whether this issue would be avoided when POLA is replaced by the proposed BRS. Additionally, the paper claims that "POLA can't differentiate through all opponent optimization steps," but more experiments are needed to support this claim. Besides, the authors also state that "the interactions between the agent and the detective mirror the foundational Stackelberg setup," but it is still not clear whether convergence to Stackelberg equilibrium helps to improve social welfare. More detailed analysis is needed to clarify these points.

2) Recently, [1] proposed a method for reasoning about players' future strategy through multiple lookahead steps. This suggests that scalability issues for POLA (or other related works) may not be a serious problem. It would be helpful if the authors could discuss some details about [1].

3) The training curves for baselines and proposed methods are missing. Also it would be helpful to test the proposed method in more complicated social dilemma environments, such as those presented in [2].

4) Some typos need to be fixed, including Alg 1., $\theta_1' \leftarrow \theta_1+z$ should be $\theta_1 \leftarrow \theta_1+z$.

**Questions:**

1) In Corollary D.3, the paper uses $r^1\left(s_t, a_t, b_t\right)+r^2\left(s_t, b_t, a_t\right)=0$ which implies $R^1(\tau)=-R^2(\tau)$. However, in proposition D.2, the authors use Lemma D.1 which assumes $r^1\left(s_t, a_t, b_t\right)=r^2\left(o_t, b_t, a_t\right)$ so as to imply  ${\mathbb{E}}\left[R^1(\tau)\right]={\mathbb{E}}\left[R^2(\tau)\right]$. Since Corollary D.3. also involves proposition D.2,This creates a potential inconsistency, and the proof of Corollary D.3 may not be correct. The authors are requested to provide clarification on this matter.






References:

[1] Recursive Reasoning in Minimax Games: A Level k Gradient Play Method

[2] Multi-agent Reinforcement Learning in Sequential Social Dilemmas

---

> ### Author Response · Authors · 2023-11-14
>
> We would like to thank the reviewer for their time and thoughtful feedback
>
> **Weaknesses:**
>
> 1. What we intended to demonstrate is that BRS produces a final, fixed policy that is not exploitable by a learning policy trained against it. We believe that the performance against MCTS demonstrates that while POLA is exploited, BRS is not. Given that both the MCTS and the learning opponent approximate the best response (but the MCTS is stronger) then we did not feel the necessity of showing the performance against a learning opponent. We do have experimental results against a learning opponent for BRS and POLA and will incorporate them in the paper. We also share the reviewer’s concern that a Stackelberg equilibrium might be undesirable for social welfare and suspect this is the fundamental reason for the need of the prosocial self-play loss. However, it would be helpful for us if the reviewer clarifies what kind of analysis would help solidify these points.
>
> 2. Regarding [1], we believe that a similar algorithm that approximates the proximal point method would be prohibitively expensive in the context of reinforcement learning. [1] scales linearly in $k$ but the underlying function w.r.t. which gradients would be computed is an approximation of the value function. This implies that to compute a single parameter update $k$ different unrollings of different policies would be required in the environment. Moreover, by the recursive structure of the update equation, these unrollings would have to be computed sequentially. This is not even considering the high variance that the chained estimators of the value function would have. Also, notice that [1] is not too different from POLA in terms of complexity.
>
> 3. We will include the training curves for these algorithms, but considering more complicated social dilemmas puts BRS to a higher standard than its predecessors, namely POLA.
>
> 4. We thank the reviewer for pointing out these typos and we will fix them.
>
> **Questions:**
>
> 1. We thank the reviewer for taking the time to read through the proof and will do our best effort to clarify it. These two statements refer to different properties of the game. In particular $r^1(s_t, a_t, b_t) = -r^2(s_t, b_t, a_t)$ refers to the zero sum property and $r^1(s_t, a_t, b_t) = r^2(o_t, b_t, a_t)$ refers to the symmetry property. The key observation is that $s_t \neq o_t$, therefore $r^1(s_t, a_t, b_t) = r^2(o_t, b_t, a_t)$ does not imply $\mathbb{E}\left[R^1(\tau)\right] =\mathbb{E}\left[R^2(\tau)\right]$. The correct implication would look something of the form  $\mathbb{E}\left[R^1(\tau)\right] =\mathbb{E}\left[R^2(\tau_o)\right]$ where for each $\tau$ there exists a $\tau_o$ with the same probability (in the self-play case). As an example consider a state $s_t$ in the Coin Game where the blue player takes the blue coin and the red player does not. The $o_t$ state refers to an analogous state in which the red player takes the red coin and the blue player does not. By symmetry of the game it should be the case that $r^1(s_t, a_t, b_t) = r^2(o_t, b_t, a_t)$. And, if the game rewards are set appropriately, it can simultaneously hold that $r^1(s_t, a_t, b_t) = -r^2(s_t, b_t, a_t)$ (that would correspond to the red player being punished with a negative reward of what the blue player got for taking the blue coin in state $s_t$).
>
> References:
>
> [1] Recursive Reasoning in Minimax Games: A Level k Gradient Play Method

---

> > ### Author Response · Authors · 2023-11-19
> >
> > We appreciate the reviewer for dedicating time to review our paper. With the discussion period nearing its conclusion, we would like to ensure that all concerns have been fully addressed. Could the reviewer please let us know if there are any additional concerns?

---

> > > ### Author Response · Authors · 2023-11-22
> > > **Regarding BRS outperforming POLA**
> > >
> > > We thank the reviewer for their time and feedback. As the reviewer requested further evidence that BRS outperforms POLA in learning reciprocation-based cooperative policies as their main concern, we would like to kindly request the reviewer's attention to the latest update of the paper with detailed analysis and comparison of POLA vs. BRS. The updated sections are noted by green as a visual cue. The mentioned sections are in appendix J (last two pages of the appendix).
> > >
> > > Here is a short summary:
> > > We analyzed the head to head rollouts between POLA seeds and BRS seeds. We observed a very interesting phenomenon:
> > > In total,  POLA breaks cooperation with BRS 950 times versus BRS's 72. BRS retaliates 84% of these times, showing stronger retaliation than POLA, which retaliates only 31% of the time. BRS cooperates initially 99% of the time, compared to POLA's 54%. This strongly indicates that BRS learns a much stronger reciprocation-based cooperation. It almost always starts with cooperation and in case of defect by opponent retaliates with high probability. POLA in the other hand just slightly prefers cooperating initially, and breaks cooperation more frequently. Also, POLA does not retaliate with high probability in rare cases in which BRS breaks the cooperation. BRS mirrors tit for tat much more closely. This indicates in a real world scenarios, one would much rather deploy a BRS agent than a POLA agent in social dilemmas.
> > >
> > > We hope our responses have addressed the reviewer's concerns. As the discussion period is almost finished, we would like to ask whether any further clarifications are needed.

---

> > > > ### Comment · Reviewer_FRpg · 2023-11-23
> > > >
> > > > I would like to thank the authors for their response and the effort they have put into enhancing the manuscript. Some of my concerns have been addressed. I am considering increasing my score from 5 to 6, but I also would like to see the feedbacks from other reviewers before making decision.

---

### Official Review · Reviewer_NynT · 2023-11-01

**Soundness:** 3 good
**Presentation:** 2 fair
**Contribution:** 2 fair
**Rating:** 6
**Confidence:** 4

**Summary:**

The authors introduce a new scalable method to tackle general-sum games called Best Response Shaping (BRS). BRS works by differentiating through an opponent that is approximating a best response. They introduce a way for the detective to condition on the agent's policy through question answering. They show promising results in the IPD and Coin Game.

**Strengths:**

Originality:

The author's proposed method is original and builds and contributes on top of prior work. The "detective" mechanism is an interesting approach to approximating the best response that I have not seen in previous literature.

Quality:

The author's evaluate in the challenging Coin Game environment and the IPD and run a large number of relevant ablatoins.

Clarity:

The authors describe the IPD, Coin Game, and related literature in depth.

Significance:

AI systems that perform well in general-sum scenarios are becoming increasingly important as AI systems become more widely-deployed.

**Weaknesses:**

Originality:

- The training paradigm described here is closely related to [PSRO](https://arxiv.org/pdf/1711.00832.pdf), which is not mentioned in the paper.

Clarity:

- The writing does not make it clear that the detective and agent take turns during optimization. The title of Algorithm 1 (that it says "a single iteration") is the *only* way I knew that this is how the training is done. Otherwise, it seems as though the detective is first trained on a pre-selected buffer B, which *significantly changes the reader's understanding of the method*.

- Section 4.1 and 4.2 really make it seem like the detective is pre-trained when it is not.

- BRS-NOSP is an important part of Figure 2, but is not defined until significantly later.

- It's still not clear which player is $\theta_1$ and which is $\theta_2$. Presumably the agent would be $\theta_1$ and the detective would be $\theta_2$, but that is flipped in Equation 4 where $\theta_r$ takes the place of $\theta_1$ but actually represents the detective and the agent is now $\theta_2$ (?)

- The whole description of "Simulation-Based Question Answering" makes the setup far more confusing than it needs to be. Here is my understanding: The conditioning vector consists of the Q-Values of each action of a random opponent policy in that state. The Q-Values are estimated by sampling trajectories. It is not described in the paper like this, and is instead treated as a complicated "Question-Answering" scheme.

- Describing and defining $\delta_A$ before describing or motivating what $Q^\text{simulation}$ was confusing as a reader, as it is uncertain how $\delta_A$ is related to anything.

- It's unclear why the authors have a paragraph about Diplomacy in the related work and a paragraph about self-play from human data. It's fine to include it, but the authors should explain, *in the paper*, why they include it.

- The authors do not write out the bi-level optimization in Equation 12, making it seem like a single-level optimization. Writing it out would make this significantly more clear that it's a bi-level process.

Quality:

- The analysis is only done on three seeds (stated in the Appendix). This is  small, considering the high variance we observe from POLA. Using more seeds for POLA because the authors "observed higher variance" is problematic from a statistical point of view. (See: Stopping rules in statistics).

- The authors state that "in this paper, [they] advocate that a reasonable point of comparison is the agent’s outcome when facing a best response opponent, which we approximate by Monte Carlo Tree Search (MCTS)." I cannot find where in the paper the authors advocate for this. This is a significant claim, since arguably POLA and LOLA are not interested in this setting as much and seem to be far more interested in learning *online* against opponents that are also *learning online*. This is rather important, as it makes head-to-head comparisons with POLA a bit more questionable, seeing as it was designed for a different setting.

Significance:

- BRS is *more exploitable than POLA* in Figure 3 against AD, seemingly contradicting the purpose of the paper. BRS-NOSP performs better, but at the cost of social welfare. However, I understand that you can't get the best of everything in general-sum games (See Q4 below).

- The authors collected head-to-head results of POLA and BRS (in the Appendix Figure 7), but have elected not to show them in the main text. It seems as though POLA outperforms BRS in the head-to-head, further indicating that BRS is exploitable.

- The authors write that the Good Shepherd is not scalable; however, their approach also seems similarly difficult to scale. (See Q3: below). What is the sample efficiency in terms of simulated environment steps?

**Questions:**

1. Why not just train directly against the MCTS opponent?

2. BRS-NORB performs near-identically to BRS in Figure 4 and is *significantly simpler*. Why not just make the method BRS-NORB?

3. Isn't calculating $\delta_A$ at every single timestep extremely expensive and not very scalable? What is the sample efficiency of BRS in terms of the total number of simulator steps?

4. Why did you run head-to-head results for the BRS variants and POLA in the Appendix, but not include them in the main text? I understand that evaluations in general-sum games can be odd, since the ultimate objective is not well-defined. I would be more convinced that BRS is valuable if there was a more explicit metric demonstrating reciprocity.

5. What is the intuition for the choice of conditioning? Why is an estimate of Q-Values of a random policy in the current state a good metric?

6. Misc: What is the reasoning for the separate optimizers? (Equation 8). Have you ablated this?

---

> ### Author Response · Authors · 2023-11-19
>
> We deeply thank the reviewer for their thorough, detailed, and in-depth feedback. Here is our response. (Sorry, we needed to cut the response to two parts)
>
> **Weaknesses**:
>
> **Originality**:
>
> We will indeed include PSRO in our literature review as it highlights the contribution of BRS over previous existing approaches. PSRO extends an existing set of policies by the best response policy to a meta-strategy of previous policies. PSRO does not differentiate through this best response generation mechanism. This is in contrast to the essence of BRS. Let’s imagine PSRO on IPD. First, it starts with a random policy. Next, always defect as the best response is added. Next, the best response to the existing set is added which is again always defect. Therefore, PSRO cannot discover tit for tat(TFT) on IPD. However, BRS is different. BRS trains an agent against a best response opponent by being aware that the opponent will be the best response. That pushes BRS to learn TFT.
>
> **Clarity**:
> 1. We thank the reviewer for their deep reading of the paper. We will rewrite the description of the algorithm to make it clear that the agent and the detective are being trained simultaneously.
> 2. We will define BRS-NOSP when we describe BRS.
>  3. The QA description shows that the detective can use any conditioning mechanism that requires querying the agent in different states. The choice for the coin game was an instance of the QA.
> 4. Diplomacy is mentioned as a large-scale game in which without forming local alliances it is impossible to win the game. Because naive RL agents don’t develop the necessary reciprocation based cooperation the successful approaches train on human data. Therefore, one long term goal is to train agents that do well on Diplomacy without any human data to teach them the tit for tat like behavior.
>
> 5. We don't have an Equation 12; perhaps you're referring to Equation 2? Indeed, we will make it clear that theta2_star represents the entire optimization process, encompassing differentiation throughout. We appreciate the reviewer's clarification on this.
>
> **Quality**:
>
> We agree that more seeds for POLA is more desirable to hinder variance. We will include more POLA seeds for the camera ready version.
>
> POLA is the only method that gives us a strong baseline for the coin game. We agree with the reviewer that while we care about generating a static policy, POLA cares more about learning online. However, the POLA authors themselves evaluate the final POLA against always defect and always cooperate opponents indicating interest in studying the properties of static final policy. Also, producing a final static policy that does well against learning opponents is valuable.
>
> **Significance**:
>
> We argue if one wants to deploy an agent in an environment full of other learning opponents maximizing their own return (naive RL) - which we believe is most cases in the real world - one would rather deploy an BRS agent than a POLA agent.
>
> To make our argument more concrete, we show BRS agents are closer to tit for tat in IPD.
>
> **First**, the best response to TFT on IPD is always cooperate. This is important as learning opponents find the best response to the deployed agent eventually. Our MCTS experiments show that while the best response to BRS is almost always cooperation, this is not the case for POLA.
>
> **Second**, TFT agents always cooperate with themselves. This is not the case for POLA while it is for BRS agents.
> **Third**, As TFT starts with cooperation at the first move it gets a lower return in head to head game with always defect. But, that is a desirable property of TFT for social settings, assuming cooperation and also keeping the door open for forgiveness. BRS agents are exploited slightly more by always defect than POLA agents. But, first, still compared to always cooperate they are exploited much less. Second, the always defect is not being a rational opponent against BRS here as it could have a much higher return if it had cooperated. Third, this tendency of BRS to break the defect loops while making it a bit more exploited against always defect gives it its ability to cooperate with itself.
>
> Regarding the head to head results, we don’t think solving social dilemmas is about comparing head to head results between two algorithms and indicating the algorithm with the higher return as the winner. TFT gets exploited by always defect but still tit for tat is a much better strategy. We can’t have complete non-exploitability and complete cooperation at the same time but we argue BRS makes a better trade off.
>
> Regarding scalability, BRS is more scalable than Good Shepherd which differentiates through a gigantic computational graph of the opponent learning from scratch to convergence. But, we agree with the reviewer, BRS needs to do the required simulations and this limits its scalability. More advanced question answering mechanisms can improve the sample efficiency.

---

> > ### Author Response · Authors · 2023-11-19
> >
> > **Questions**:
> >
> > 1. This was our initial idea. But, we ask the reviewer’s attention to the fact that we need to differentiate through the learning mechanism of the MCTS and we need to treat MCTS as a black box. An algorithm that query’s the agent on a bunch of states, receives the action, and outputs a number: the return of the agent against the MCTS. The differentiation is therefore done with REINFORCE. It worked for us in a 2x1 board for the coin game. However, as the board size increases we needed near one thousand MCTS samples to approximate that best response. This induces a huge variance in the REINFORCE signal therefore rendering differentiation not possible. Consequently, we designed the detective. An agent that approximates the best response in a differentiable computational graph so the optimization function can be differentiated directly and not via REINFORCE estimators.
> >
> > 2. The replay buffer improves the results a little bit. We think it is a trade-off between higher performance and algorithm complexity. We agree with the reviewer that this trade off may be worth it.
> >
> > 3. We agree with the reviewer that this is computationally expensive. However, we believe it is less expensive than unrolling explicit optimization computational graphs. But, we agree BRS is not the ultimate solution to scalable solving of social dilemmas. It is a stepping stone to look at the problem from a different direction than unrolling explicit optimization steps (as other methods like LOLA and POLA do.)
> >
> > 4. Please see the section in our response to significance regarding head to head results. In summary, we don’t think head to head results of two algorithms correctly convey solving a social dilemma.
> >
> > 5. This is a good indicator whether an agent retaliates if you defect against it. Assume the opponent is near a coin. If the random opponent’s return drops by picking the agent coin here, it indicates the agent is retaliatory. If it does not, it indicates the agent is unconditional cooperative.
> >
> > 6. The gradient through the detective has higher variance. Summing up two signals of different variance usually leads to one overpowering the other specially in presence of momentum in optimizers. We ablated this and found separate optimizers for each term to work more robustly.
> >
> >
> > We deeply appreciate this review. The raised questions are excellent and incorporating this feedback in the main paper will significantly improve the paper's clarity. We hope the provided clarification points addressed the reviewer's questions.

---

> > > ### Comment · Reviewer_NynT · 2023-11-21
> > > **Thank you for the detailed response**
> > >
> > > I would like to thank the authors for their detailed response and updated manuscript. Their clarifying responses have helped me understand the method much better. It would be good if the authors could incorporate some of this discussion into the paper (e.g. the intuition in Q5 or their argument about why they "would rather deploy an BRS agent than a POLA agent")
> > >
> > > I believe the achieved results are impressive and the method is interesting enough to warrant an accept. I have updated my score accordingly.
> > >
> > > If I were to update the score further, it would be good if the authors could have a more detailed analysis of the return in Coin Game. For example, could they have an explicit measure of reciprocity in Coin Game so that we know whether BRS is being exploited by AD / POLA or if it is just reciprocating? Could we get a table reporting the head-to-head results? I understand that an individual head-to-head result is not informative, but I think such a table would make it very clear which algorithm is better on "average" and which algorithm is the best in each situation. e.g. Tit-for-tat would indeed get slightly exploited by Always Defect, but it would generally be one of the best performing algorithms overall.

---

> > > > ### Author Response · Authors · 2023-11-22
> > > > **Thank you for the great suggestion**
> > > >
> > > > We would like to thank the reviewer for suggesting that we delve deeper into the analysis of the rollouts between the POLA seeds and BRS seeds. We found empirical evidence that BRS results in strongly reciprocation-based cooperative agents compared to POLA. We have added a detailed discussion of the results in the appendix. Here, we
> > > > provide a short summary:
> > > >
> > > > **League results:**
> > > >
> > > >  We have added a head to head result averaged over 32 rollouts for each POLA seed against each BRS seed. First, POLA seeds vary in their behaviour, while BRS seeds are robustly showing the same behaviour. We can roughly categorized the POLA seeds to two categories: 1) cooperative but exploitable by MCTS (and other POLA seeds) 2) defective but not cooperative with themselves or the MCTS (exploiting other POLA seeds) . In contrast to all BRS seeds, there is no single POLA seed in which the best response, approximated by the MCTS agent, is always cooperate. Also, BRS seeds always cooperate within themselves.
> > > >
> > > > **Head to head behaviour analysis:**
> > > >
> > > > The most interesting result, is the analysis of retaliation behaviours in head to head runs between POLA and BRS seeds. In the coin game, let us call every time an agents lets the opponent to pick a coin that has the opponent color as the agent  cooperating with the opponent. Also, let us call every time an agent picks up the opponent coin as the agent defecting against the opponent.
> > > >
> > > > We observed that in total 950 times POLA breaks the cooperation with BRS while BRS cooperated last time. BRS only breaks the cooperation 72 times. However, while BRS retaliates after 798 (84%) of those 950 times indicating strong retaliatory behaviour, POLA only retaliates after 22 (31%) of those 72 times indicating weak retaliatory behaviour. We suspect that is the reason that MCTS can exploit POLA. POLA agents do not retaliate strongly.  Also, POLA starts 54% of the time with cooperation while BRS cooperates initially 99% of the time. We believe this empirical analysis further strengthens our confidence that BRS learned a stronger reciprocation-based cooperative policy compared to POLA.
> > > >
> > > > We would like to first thank the reviewer for their active and swift engagement in the discussion period. Their feedback improved our experiments and our paper strongly and made our contributions clear. As the discussion period is almost ending,  we kindly ask the reviewer if there are any further concerns

---

### Official Review · Reviewer_Hmjc · 2023-11-01

**Soundness:** 3 good
**Presentation:** 4 excellent
**Contribution:** 2 fair
**Rating:** 5
**Confidence:** 4

**Summary:**

The paper introduces a new opponent modelling tool which differentiates through a constructed opponent that approximates the
agent's best response policy in a best-response algorithm-like manner. The paper include various empirical evaluations to validate the proposed mechanisms.

**Strengths:**

The paper introduces a neat method for overcoming some of the weaknesses in current opponent modelling approaches. The paper is both well-written and well structured with good empirical results to support the authors' claims.

**Weaknesses:**

**Missing references**

Some important references have been missed such as [1] and [2].
For example [1] introduces a method for incorporating diversity into best-response dynamics leading to a diverse fictitious play/diverse best response dynamics. Since fictitious play is based on a type of best-response method, a discussion on how the authors’ paper compares to [1] is needed to be able to properly evaluate the authors’ current contribution.

**Limitations**
It seems that method requires the agent to have access to the detectives received rewards (and vice-versa) – this seems to limit the application to various non-cooperative multi-agent settings where at least the reward may be privately received.

In the abstract, the authors state the focus of the paper is in partially-competitive games, however the study of the cooperation regularization method does not shed any light on games that may be of mixed character (for example games in which $r^1(s,\cdot)=-r^2(s,\cdot)$ and $r^1(s',\cdot)=r^2(s',\cdot)+k$ for $s\neq s'$).

In the conclusion, the authors state that one of the aims is improving the scalability and non-exploitability of agents. However, it is unclear about the practicalities of such a method in larger scale games as well as how it could ever be adapted beyond two-player games.



[1] Perez-Nieves, Nicolas, et al. "Modelling behavioural diversity for learning in open-ended games." International conference on machine learning. PMLR, 2021.
[2] Yang, Yaodong, et al. "Multi-agent determinantal q-learning." International Conference on Machine Learning. PMLR, 2020.

**Questions:**

Q1: How does the method compare to other approaches that introduce diversity into opponent modelling such as [1] and [2]?

Q2: Can the authors comment on whether or not (and how) this could be applied in games where the opponent’s rewards are unobserved?

Q3: Although the cooperation regularization method yields a useful tool that does not produce cooperative effects in zero-sum games, its effect in games in which some game states the players ought to behave adversarially and other game states cooperatively (for example the weakest link) is unclear to me. Can the authors shed any light on this?

[1] Perez-Nieves, Nicolas, et al. "Modelling behavioural diversity for learning in open-ended games." International conference on machine learning. PMLR, 2021.
[2] Yang, Yaodong, et al. "Multi-agent determinantal q-learning." International Conference on Machine Learning. PMLR, 2020.

---

> ### Author Response · Authors · 2023-11-14
>
> We would like to thank the reviewer for their time and thoughtful feedback
>
> **Weaknesses:**
>
> **Missing References**
>
> We appreciate the reviewer pointing these references out, we will include them in the literature review.  We would like to clarify that [1] makes two major assumptions that significantly simplify the problem. First, [1] is restricted to normal form games so the best response that is being computed corresponds to a mixed strategy. In contrast, BRS computes an approximation to the best response in policy space, which is significantly harder (and unclear how to do using [1]). Second, because [1] deals with matrix form games, it is relatively straightforward to define a sampling method for the policies, in their case a Determinantal Point Process, due to the linear nature of the underlying strategies. In contrast BRS adds diversity to fictitious play by keeping a replay buffer of encountered agents and adding noise to the policy parameters. So it remains unclear to us how a) [1] is able to scale beyond normal form games into games that require a neural network policy parameterization and b) that [1] provides a solution to social dilemmas.
>
> **Limitations**:
>
>  Our method assumes that the rewards for all agents are visible but we would like to remind the reviewer that this kind of assumptions are normal in the literature: POLA makes this assumption and LOLA makes a stronger one (assuming access to policy parameters). The reviewer correctly points out that this limits the application of BRS to non-cooperative settings where the reward may be privately received.
>
> What we intended to say by writing “partially-competitive games” was social dilemmas so we will probably change all the appearances of this term to clarify the scope of our paper. We are not providing a solution for zero sum or fully cooperative games.
>
> Regarding the conclusion, the reviewer correctly points out that BRS is not a method that works for n-player games as it is stated in the limitations of the paper. However, because BRS allows for the use of neural network policies it can scale to more complex games that use higher dimensional representations of the environment.
>
> **Questions**
>
> 1. In BRS since the policies are non-linear, diversity is found using a replay buffer of policies and adding gaussian noise to the parameters of these policies. Which contrasts with the Determinantal Point Processes used in [1] and [2]. However, diversity seems to be not necessary for the Coin Game experiments as shown in the ablation section.
>
> 2. BRS cannot be applied to games where the reward for the other player remains unobserved unless that reward can be inferred from the behavior of the other player. Nonetheless, similar assumptions are done in the literature and used only during training; at deployment time the assumption is not necessary.
>
> 3. We do not know the answer to this question but we believe it is a very important one that should be explored in future work. Weakest link resembles Pure Diplomacy in the necessity for cooperation during the initial stages of the game and the need for competitive policies afterwards, and Pure Diplomacy has been one of the setups that we are interested in pursuing. We also believe BRS may work in a two player game with this structure, because the first gradient component (against the detective) induces adversarial behavior whereas the second one (self-play) induces cooperative behavior.
>
> Under these considerations, would you be willing to increase your score?
>
> References:
>
> [1] Perez-Nieves, Nicolas, et al. "Modelling behavioural diversity for learning in open-ended games." International conference on machine learning. PMLR, 2021.
>
> [2] Yang, Yaodong, et al. "Multi-agent determinantal q-learning." International Conference on Machine Learning. PMLR, 2020.

---

> > ### Author Response · Authors · 2023-11-19
> >
> > We thank the reviewer for their time on reviewing our paper. As the discussion period nears its conclusion, we aim to address every concern thoroughly. Could the reviewer indicate if any further comments or unresolved issues remain?

---

### Author Response · Authors · 2023-11-21
**Paper Updated**

Dear Reviewers,

Thank you once more for your efforts and valuable insights during the review process. We believe we have comprehensively responded to your inquiries and revised our manuscript to include your suggestions. This includes fixing typos, adding the suggested related works, clarifications and training curves. Also, the changes to the main text are shown in green colour as a visual cue. As the discussion period is nearing its end please let us know if any aspect requires further clarification.

---

### Author Response · Authors · 2023-11-22
**Paper Updated with Incorporated Feedbacks and Detailed Analysis of POLA vs. BRS Retaliation Behaviours Indicating that BRS Retaliates Significantly Stronger than POLA**

We would like to thank all the reviewers for their time and feedback.

We have incorporated several suggestions that came out during the rebuttal period. First, we have corrected several typos pointed out by the reviewers and clarified certain sections of the notation that were not clear.

We have enriched our literature review by discussing the following references:
   1. Perez-Nieves, Nicolas, et al. "Modelling behavioural diversity for learning in open-ended games." International conference on machine learning. PMLR, 2021. as suggested by reviewer Hmjc
 2. Yang, Yaodong, et al. "Multi-agent determinantal q-learning." International Conference on Machine Learning. PMLR, 2020. as suggested by reviewer Hmjc
 3. Leibo, Joel, et al. “Multi-agent Reinforcement Learning in Sequential Social Dilemmas.” 2017 FRpg
 4. Lanctot, Marc, et al. "A unified game-theoretic approach to multiagent reinforcement learning." Advances in neural information processing systems 30 (2017). as suggested by reviewer NynT

We have also added the following figures:
1.  Training curves for BSR as requested by reviewer FRpg.
2.  Head-to-head results between BSR and POLA as requested by reviewer NynT.

We have added detailed analysis:
1. Adding details of head to head POLA vs. BRS results as suggested by reviewer NyNT: **BRS learns stronger reciprocation-based cooperative agents than POLA**. POLA breaks cooperation with BRS 950 times versus BRS's 72. **BRS retaliates 84% of these times**, showing stronger retaliation than **POLA, which retaliates only 31%** of the time. BRS cooperates initially 99% of the time, compared to POLA's 54%.

2.  Adding explanation of why one would rather deploy BRS than POLA as a result of discussion with reviewer NyNT: BRS mirrors TFT unlike POLA.

As the discussion period is almost ended, we would like to thank the reviewers for their insightful feedback and kindly ask if there is any remaining concern.

---

### Meta-Review · Area_Chair_1qDy · 2023-12-06

**Metareview:**

There were concerns with the technical correctness of part of the paper.  In addition, several reviewers mentioned concerns with generalizaility and scalabiity of the approach (e.g. beyond two players).

I do agree with the authors that the Coin game is interesting and we don't have really good solutions for it.

I disagree with the authors about the meaning of the term "self play". While some people use it in a way that entails parameter sharing, it is also very common to use the term 'self play' to refer to any co-training algorithm, regardless of whether agents share parameters or not. It's the kind of term you always have to precisely define since people use it differently from one another.

**Justification For Why Not Higher Score:**

I think the points raised by the reviewers about how parameter sharing limits the impact of this work are likely correct. I note also that even if we removed the score of the reviewer who game the lowest score then the average would still only be 5.3, which is likely not enough to accept it.

**Justification For Why Not Lower Score:**

N/A

---

### Decision · Program_Chairs · 2024-01-16

Reject